# STEIN SELF-REPULSIVE DYNAMICS: BENEFITS FROM PAST SAMPLES

## ABSTRACT

We propose a new Stein self-repulsive dynamics for obtaining diversified samples from intractable un-normalized distributions. Our idea is to introduce Stein variational gradient as a repulsive force to push the samples of Langevin dynamics away from the past trajectories. This simple idea allows us to significantly decrease the auto-correlation in Langevin dynamics and hence increase the effective sample size. Importantly, as we establish in our theoretical analysis, the asymptotic stationary distribution remains correct even with the addition of the repulsive force, thanks to the special properties of the Stein variational gradient. We perform extensive empirical studies of our new algorithm, showing that our method yields much higher sample efficiency and better uncertainty estimation than vanilla Langevin dynamics.

## 1 INTRODUCTION

Drawing samples from complex un-normalized distributions is one of the most basic problems in statistics and machine learning, with broad applications to enormous research fields that rely on probabilistic modeling. Over the past decades, large amounts of methods have been proposed for approximate sampling, including both Markov Chain Monte Carlo (MCMC) (e.g., Brooks et al., 2011) and variational inference (e.g., Wainwright et al., 2008).

MCMC works by simulating Markov chains whose stationary distributions match the distributions of interest. Despite nice asymptotic theoretical properties, MCMC is widely criticized for its slow convergence rate in practice. In difficult problems, the samples drawn from MCMC are often found to have high auto-correlation across time, meaning that the Markov chains explore very slowly in the configuration space. When this happens, the samples returned by MCMC only approximate a small local region, and under-estimate the probability of the regions un-explored by the chain.

Stein variational gradient descent (SVGD) (Liu & Wang, 2016) is another type of approximation sampling methods designed to overcome the limitation of MCMC. Instead of drawing random samples sequentially, SVGD evolves a pre-defined number of particles (or sample points) in parallel with a special interacting particle system to match the distribution of interest by minimizing the KL divergence. In SVGD, the particles interact with each other to simultaneously move towards the high probability regions following the gradient direction, and also move away from each other due to a special repulsive force. As a result, SVGD allows us to obtain diversified samples that correctly represent the variation of the distribution of interest. SVGD has been found a promising tool for solving difficult sampling problems in which diversity promotion is critical (e.g., Feng et al., 2017; Haarnoja et al., 2017; Pu et al., 2017; Liu et al., 2017; Gong et al., 2019). Various extensions have been developed (e.g., Han & Liu, 2018; Chen et al., 2018; Liu et al., 2019; Wang et al., 2019a).

However, one problem of SVGD is that it theoretically requires to run an infinite number of chains in parallel in order to approximate the target distribution asymptotically (Liu, 2017). With a finite number of particles, the fixed point of SVGD does still provide a *prioritized, partial* approximation to the distribution in terms of the expectation of a special case of functions (Liu & Wang, 2018). Nevertheless, it is still desirable to develop a variant of "single-chain SVGD", which only requires to run a single chain sequentially like MCMC to achieve the correct stationary distribution asymptotically in time, with no need to take the limit of infinite number of parallel particles.

In this work, we propose an example of *single-chain SVGD* by integrating the special repulsive mechanism of SVGD with gradient-based MCMC such as Langevin dynamics. Our idea is to use repulsive term of SVGD to enforce the samples in MCMC away from the past samples visited at previous iterations. Such a new *self-repulsive dynamics* allows us to decrease the auto-correlation in MCMC and hence increase the mixing rate, but still obtain the same stationary distribution thanks to the special property of the SVGD repulsive mechanism.

We provide throughout theoretically analysis of our new method, establishing it asymptotic convergence to the target distribution. As we show in the work, the analysis is highly non-trivial, because our new self-repulsive dynamics is a non-linear, high-order Markov process. Empirically, we extensively evaluate our methods on an array of challenging sampling tasks, showing that our method yields much better uncertainty estimation and larger effective sample size.

## 2 BACKGROUND: LANGEVIN DYNAMICS AND SVGD

In this section, we give a brief introduction on Langevin dynamics (Rossky et al., 1978) and Stein Variational Gradient Descent (SVGD) (Liu & Wang, 2016), which we integrate teogehter to develop our new self-repulsive dynamics for more efficient sampling.

**Langevin dynamics**   Langevin dynamics is a basic gradient based MCMC method. For some target distribution on $\mathbb{R}^d$ with density function $\rho^*(\boldsymbol{\theta}) \propto \exp(-V(\boldsymbol{\theta}))$, where $V \colon \mathbb{R}^d \mapsto \mathbb{R}$ is the potential function, the (Euler-discrerized) Langevin dynamics simulates a Markov chain with the following rule:

$$\boldsymbol{\theta}_{k+1} = \boldsymbol{\theta}_k - \eta \nabla V(\boldsymbol{\theta}_k) + \sqrt{2\eta}\boldsymbol{e}_k, \qquad \boldsymbol{e}_k \sim \mathcal{N}(0, \mathbf{I}),$$

where $k$ denotes the number of iterations, $\{\boldsymbol{e}_k\}$ are independent standard Gaussian noise, and $\eta$ is a step size parameter. It is well known that the limiting distribution of $\boldsymbol{\theta}_k$ when $k \to \infty$ approximates the target distribution when $\eta$ is sufficiently small.

Because the updates in Langevin dynamics are local and incremental, new points generated by the dynamics is highly correlated to the past sample. As a result, we need to run Langevin dynamics sufficiently long in order to obtain diverse samples.

**Stein Variatinal Gradient Descent (SVGD)**   Different from Langevin dynamics, SVGD iteratively evolves a pre-defined number of particles in parallel. Starting from an initial set of particles $\{\boldsymbol{\theta}_0^i \colon i = 1, ..., M\}$, SVGD updates the $M$ particles in parallel by

$$\boldsymbol{\theta}_{k+1}^i = \boldsymbol{\theta}_k^i + \eta \boldsymbol{g}(\boldsymbol{\theta}_k^i; \; \hat{\delta}_k^M) \quad \forall i = 1, \dots, M,$$

where the velocity field, which we denote by $\boldsymbol{g}(\boldsymbol{\theta}_k^i; \; \hat{\delta}_k^M)$, depends the empirical distribution of the current set of particles $\hat{\delta}_k^M := \frac{1}{M} \sum_{j=1}^{M} \delta_{\boldsymbol{\theta}_k^j}$ in the following way,

$$\boldsymbol{g}(\boldsymbol{\theta}_k^i; \; \hat{\delta}_k^M) = \mathbb{E}_{\boldsymbol{\theta} \sim \hat{\delta}_k^M} \left[ \underbrace{-K(\boldsymbol{\theta}, \boldsymbol{\theta}_k^i)\nabla V(\boldsymbol{\theta})}_{\text{Confining Term}} + \underbrace{\nabla_{\boldsymbol{\theta}} K(\boldsymbol{\theta}, \boldsymbol{\theta}_k^i)}_{\text{Repulsive Term}} \right].$$

Here $\delta_{\boldsymbol{\theta}}$ is the Dirac measure centered at $\boldsymbol{\theta}$, and hence $\mathbb{E}_{\boldsymbol{\theta} \sim \hat{\delta}_k^M}[\cdot]$ denotes averaging on the particles. The $K(\cdot, \cdot)$ is a positive definite kernel, such as the Gaussian RBF kernel, specified by users.

Note that $\boldsymbol{g}(\boldsymbol{\theta}_k^i; \; \hat{\delta}_k^M)$ consists of a confining term and repulsive term: the confining term pushes particles to move towards high density region, and the repulsive term prevents the particles from colliding with each other. It is the balance of these two terms that allows us to asymptotically approximate the target distribution $\rho^*(\boldsymbol{\theta}) \propto \exp(-V(\boldsymbol{\theta}))$ at the fixed point, when the number of particles goes to infinite. We refer the readers to Liu & Wang (2016); Liu (2017); Liu & Wang (2018) for throughout theoretical justifications of SVGD. But a quick, informal way to justify the SVGD update is through the *Stein's identity*, which shows that the velocity field $\boldsymbol{g}(\boldsymbol{\theta}; \; \rho)$ equals zero when $\rho$ equals the true distribution $\rho^*$, that is,

$$\boldsymbol{g}(\boldsymbol{\theta}'; \rho^*) = \mathbb{E}_{\boldsymbol{\theta} \sim \rho^*} \left[ -K(\boldsymbol{\theta}, \boldsymbol{\theta}')\nabla V(\boldsymbol{\theta}) + \nabla_{\boldsymbol{\theta}} K(\boldsymbol{\theta}, \boldsymbol{\theta}') \right] = 0, \quad \forall \boldsymbol{\theta}' \in \mathbb{R}^d. \tag{1}$$

This shows that SVGD would converge if the particle distribution already forms a closed approximation to the target distribution $p$, meaning that the target distributions forms an (approximate) fixed point of the update.

# 3 STEIN SELF-REPULSIVE DYNAMICS

In this work, we propose to integrate Langevin dynamics and SVGD to simultaneously decrease the auto-correlation of Langevin dynamics and eliminate the need for running parallel chains in SVGD. The idea is to use Stein repulsive force between the the current sample and the past samples, hence forming a new self-avoiding dynamics with fast convergence speed.

Specifically, assume we run a single Markov chain like Langevin dynamics, where $\boldsymbol{\theta}_k$ denotes the particle at the $k$-th iteration. Denote by $\tilde{\delta}_k^M$ the empirical distribution of $M$ samples taken from the past iterations, i.e.,

$$\tilde{\delta}_k^M := \frac{1}{M} \sum_{j=1}^{M} \delta_{\boldsymbol{\theta}_{k-jc_\eta}}, \qquad c_\eta = c/\eta,$$

where $c_\eta$ is a thinning factor, which scales inversely with the step size $\eta$, introduced to slim the sequence of past samples. Compared with the $\hat{\delta}_k^M$ in SVGD, which is averaged over $M$ parallel particles, $\tilde{\delta}_k^M$ is averaged across time over $M$ past samples. Given this, our Stein self-repulsive dynamics updates the sample via

$$\boldsymbol{\theta}_{k+1} \leftarrow \boldsymbol{\theta}_k + \underbrace{(-\eta V(\boldsymbol{\theta}_k) + \sqrt{2\eta}\boldsymbol{e}_k)}_{\text{Langevin}} + \underbrace{\eta\alpha\boldsymbol{g}(\boldsymbol{\theta}_k; \tilde{\delta}_k^M)}_{\text{Stein Repulsive}}, \tag{2}$$

in which the particle is updated with the typical Langevin gradient, plus a Stein repulsive force against the samples from the previous iterations. $\alpha \geq 0$ is a parameter that controls the magnitude of the Stein repulsive term. In this way, the particles are pushed away from the past samples, and hence admits lower auto-correlation and faster convergence speed. Importantly, the addition of the repulsive force *does not impact* the asymptotic stationary distribution, thanks to Stein's identity in (1). This is because when if the self-repulsive dynamics have converged to the target distribution $\rho^*$, such that $\theta_k \sim \rho^*$ for all $k$, the Stein self-repulsive term would equal to zero in expectation due to Stein's identity and hence does not introduce additional bias over Langevin dynamics. Rigorous theoretical analysis of this idea is developed in Section 4.

**Practical Algorithm**    Because $\tilde{\delta}_k^M$ is averaged across the past samples, it is necessary to introduce a burn-in phase with the repulsive dynamics. Therefore, our overall procedure works as follows,

$$\boldsymbol{\theta}_{k+1} = \begin{cases} \boldsymbol{\theta}_k - \eta\nabla V(\boldsymbol{\theta}_k) + \sqrt{2\eta}\boldsymbol{e}_k, & k < Mc_\eta \\ \boldsymbol{\theta}_k + \eta\left[-\nabla V(\boldsymbol{\theta}_k) + \alpha\boldsymbol{g}(\boldsymbol{\theta}_k; \tilde{\delta}_k^M)\right] + \sqrt{2\eta}\boldsymbol{e}_k, & k \geq Mc_\eta. \end{cases} \tag{3}$$

It includes two phases. The first phase is the same as the Langevin dynamics which collects the initial $M$ samples used in the second phase while serves as a warm start. The repulsive gradient update is introduced in the second phase to encourage the dynamics to visit the under-explored density region. We call this particular instance of our algorithm Self-Repulsive Langevin dynamics (SRLD), self-repulsive variants of more general dynamics is discussed in Section 5.

**Remark**    Notice that, the first phase is introduced to collect the initial $M$ samples. However, it's not really necessary to generate the initial $M$ samples with Langevin dynamics. We can simply use some other initialization distribution and get $M$ initial samples from that distribution. In practice, we find using Langevin dynamics to collect the initial samples is natural and it can also be viewed as the burn-in phase before sampling, so we use (3) in all of the other experiments.

**Remark**    The general idea of introducing self-repulsive terms inside MCMC or other iterative algorithms is not new itself. For example, in molecular dynamics simulations, an algorithm called metadynamics (Laio & Parrinello, 2002) has been widely used, in which the particles are repelled away from the past samples in a way similar to our method, but with a typical repulsive function, such as $\sum_j D(\theta_k, \theta_{k-jc_\eta})$, where $D(\cdot, \cdot)$ is any notation of dis-similarity. However, introducing

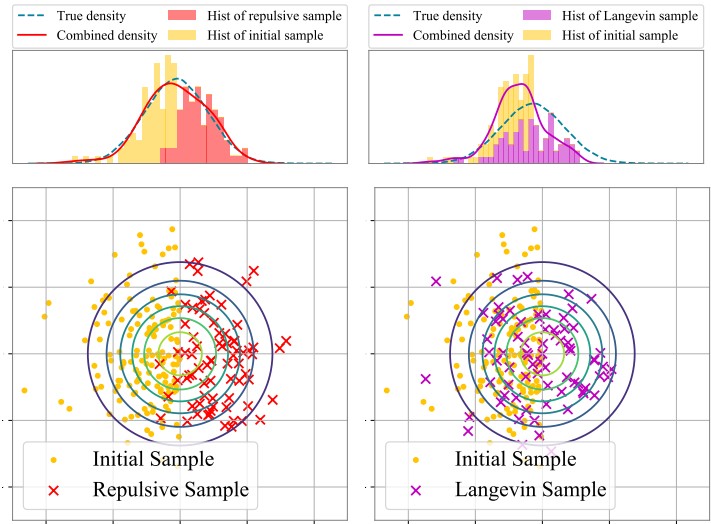

Figure 1: An illustrative example showing the advantage of our Self-Repulsive Langevin dynamics. With a set of initial examples locating on the left part of the target distribution (show in yellow), Self-Repulsive Langevin dynamics is forced to explore the right part more frequently, yielding an accurate approximation when combined with the initial samples. Langevin dynamics, however, does not take the past samples into account and yields a poor overall approximation.

an arbitrary repulsive force would alter the stationary distribution of the dynamics, introducing a harmful bias into the algorithm. The key highlight of our approach, as reflected by our theoretical results in Section 4, is the unique property of the Stein repulsive term, that allows us to obtain the correct stationary distribution even with the addition of the repulsive force.

**Remark**    Recently, (Zhang et al., 2018) proposed a different combination of SVGD and Langevin dynamics, in which the Langevin force is directly added to a set of particles that evolve in parallel with SVGD. Using our terminology, their system is

$$\boldsymbol{\theta}_{k+1}^i = \boldsymbol{\theta}_k^i + (-\eta V(\boldsymbol{\theta}_k^i) + \sqrt{2\eta}\boldsymbol{e}_k^i) \; + \; \eta\alpha\boldsymbol{g}(\boldsymbol{\theta}_k^i;\, \hat{\delta}_k^M), \quad \boldsymbol{e}_k \sim \mathcal{N}(0, \mathbf{I}) \quad \forall i = 1, \dots, M.$$

This is significantly different from our method on both motivation and practical algorithm. Their algorithm still requires to simulate $M$ chains of particles in parallel like SVGD, and was proposed to obtain easier theoretical analysis than the deterministic dynamics of SVGD. Our work is instead motivated by the practical need of decreasing the auto-correlation in Langevin dynamics, and avoiding the need of running multiple chains in SVGD, and hence must be based on *self-repulsion* against past samples along a single chain.

In another recent work, (Chen et al., 2018) proposed a $\pi$-SGLD method, which simulates the linear combination of the evalutionary partial differential equations of SVGD and Langevin dynamics using discrete gradient flow with blob-based method. Their method is again motivated by the theoretical interest of discovering new categories of algorithms, and does not involve self-repulsive on a single chain like our method.

**An Illustrative Example**    Here we give an illustrative example to show the key advantage of our self-repulsive dynamics. Assume that we want to sample from a bi-variate Normal distribution shown in Figure 1. Unlike standard settings, we assume that we have already obtained some initial samples (yellow dots in Figure 1) before running the dynamics. The initial samples are assumed to concentrate on the left part of the target distribution as shown in Figure 1. In this extreme case, since the left part of the distribution is over-explored by the initial samples, it is desirable to have the subsequent new samples to concentrate more on the un-explored part of the distribution. However, standard Langevin dynamics does not take this into account, and hence yielding a bias overall representation of the true distribution (see the left panel). With our self-repulsive dynamics, the new

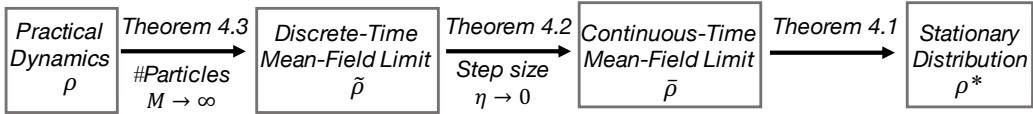

Figure 2: Roadmap of the theoretical analysis. Theorem 4.3 shows the mean-field limit when $M \to \infty$. Theorem 4.2 bounds the time discretization error. And Theorem 4.1 shows that the limiting distribution of the continuous-time mean field dynamics is the target distribution we want.

samples are forced to explore the un-explored region more frequently, allowing us to obtain a much more accurate approximation when combining the new and initial samples.

## 4 THEORETICAL ANALYSIS OF STEIN SELF-REPULSIVE DYNAMICS

In this section, we provide theoretical analysis of the self-repulsive dynamics. We establish that our self-repulsive dynamics converges to the correct target distribution asymptotically, in the limit when $M$ approaches to infinite and the step size $\eta$ approaches to 0. This is a highly non-trivial task, because the self-repulsive dynamics is a highly complex, non-linear and high order Markov stochastic process. We attack this problem by breaking the proof into the following three steps illustrated in Figure 2:

1) At the limit of large particle sizes $M \to \infty$ (called the **mean field limit**), we show that practical dynamics in (3) is closely approximated by a **discrete-time mean-field dynamics** characterized by (4) below.

2) By further taking the limit of small step size $\eta \to 0^+$ (called the **continuous-time limit**), the dynamics in (4) converges to a **continuous-time mean-field dynamics** characterized by (5) below.

3) We show that the dynamics in (5) converges to the target distribution.

**Remark** As we mentioned in Section 3, we introduce the first phase to collect the initial $M$ samples for the second phase, and our theoretical analysis follows this setting to make our theory as close to the practice. However, the theoretical analysis can be generalized to the setting of drawing $M$ initial samples from some initialization distribution with almost identical argument.

**Notations** We use $\|\cdot\|$ and $\langle \cdot, \cdot \rangle$ to represent the $\ell_2$ vector norm and inner product respectively. The Lipschitz norm and bounded Lipschitz norm of a function $f$ are defined by $\|f\|_{\mathrm{Lip}}$ and $\|f\|_{\mathrm{BL}}$. The KL divergence, Wasserstein-2 distance and Bounded Lipschitz distance between distribution $\rho_1, \rho_2$ are denoted as $\mathbb{D}_{\mathrm{KL}}[\rho_1 \| \rho_2]$, $\mathbb{W}_2[\rho_1, \rho_2]$ and $\mathbb{D}_{\mathrm{BL}}[\rho_1, \rho_2]$ respectively.

### 4.1 MEAN-FIELD AND CONTINUOUS-TIME LIMITS

To fix the notation, we denote by $\rho_k := \mathrm{Law}(\boldsymbol{\theta}_k)$ the distribution of $\boldsymbol{\theta}_k$ at time $k$ of the practical self-repulsive dynamics (3), which we refer as the **practical dynamics** in the sequel, when the initial particle $\boldsymbol{\theta}_0$ is drawn from an initial continuous distribution $\rho_0$ supported on $\mathbb{R}^d$. Note that given $\rho_0$, the subsequent $\rho_k$ can be recursively defined through dynamics (3). Due to the diffusion noise in Langevin dynamics, all $\rho_k$ are continuous distributions supported on $\mathbb{R}^d$. We now introduce the limit dynamics when we take the mean-field limit ($M \to +\infty$) and then the continuous-time limit ($\eta \to 0^+$).

**Discrete-Time Mean-Field Dynamics ($M \to +\infty$)** In the limit of $M \to \infty$, we show that our practical dynamics is closely approximated by the following dynamics, in which the empirical measures are replaced by the continuous distributions of the particles themselves.

$$
\tilde{\boldsymbol{\theta}}_{k+1} = \begin{cases} \tilde{\boldsymbol{\theta}}_k - \eta \nabla V(\tilde{\boldsymbol{\theta}}_k) + \sqrt{2\eta} \boldsymbol{e}_k, & k \leq Mc_\eta \\ \tilde{\boldsymbol{\theta}}_k + \eta \left[ -\nabla V(\tilde{\boldsymbol{\theta}}_k) + \alpha \boldsymbol{g}(\boldsymbol{\theta}_k, \tilde{\rho}_k^M) \right] + \sqrt{2\eta} \boldsymbol{e}_k, & k \geq Mc_\eta \end{cases} \tag{4}
$$

where $\tilde{\rho}_k^M = \frac{1}{M} \sum_{j=1}^M \tilde{\rho}_{k-jc_\eta}$ by $\tilde{\rho}_k := \mathrm{Law}(\tilde{\boldsymbol{\theta}}_k)$ the (smooth) distribution of $\tilde{\boldsymbol{\theta}}_k$ at time-step $k$ when the dynamics is initialized with $\tilde{\boldsymbol{\theta}}_0 \sim \tilde{\rho}_0 = \rho_0$. Compared with the practical dynamics

in (3), the difference is that the empirical distribution $\tilde{\delta}_k^M$ is replaced by the smooth distribution $\tilde{\rho}_k^M$. Similar to the recursive definition of $\rho_k$ following dynamics (3), $\tilde{\rho}_k$ is also recursively defined through dynamics (4), starting from $\tilde{\rho}_0 = \rho_0$. As we show in Theorem 4.3, if the auto-correlation of $\boldsymbol{\theta}_k$ decays fast enough and $M$ is sufficiently large, $\tilde{\rho}_k^M$ is well approximated by the empirical distribution $\tilde{\delta}_k^M$ in (3), and further the two dynamics ((3) and (4)) converges to each other in the sense that $\mathbb{W}_2[\rho_k, \tilde{\rho}_k] \to 0$ as $M \to \infty$ for any $k$. Note that in taking the limit of $M$, we need to ensure that we run the dynamics for more than $Mc_\eta$ steps. Otherwise, SRLD degenerates to Langevin dynamics as we stop the chain before we collect $M$ samples.

**Continuous-Time Mean-Field Dynamics ($\eta \to 0^+$)**   In the limit of zero step size ($\eta \to 0^+$), the discrete-time mean field dynamics in (4) can be shown to converge to the following continuous-time mean-field dynamics:

$$d\bar{\boldsymbol{\theta}}_t = \begin{cases} -\nabla V(\bar{\boldsymbol{\theta}}_t)dt + d\mathcal{B}_t, & t \in [0, Mc) \\ \left[ -\nabla V(\bar{\boldsymbol{\theta}}_t) + \alpha \boldsymbol{g}(\boldsymbol{\theta}_k, \ \bar{\rho}_t^M) \right] dt + d\mathcal{B}_t, & t \geq Mc, \end{cases} \tag{5}$$

where $\bar{\rho}_t^M := \frac{1}{M} \sum_{j=1}^M \bar{\rho}_{t-jc}(\cdot)$ and $\bar{\rho}_t = \text{Law}\left(\bar{\boldsymbol{\theta}}_t\right)$ is the distribution of $\bar{\boldsymbol{\theta}}_t$ at a continuous time point $t$ with $\boldsymbol{\theta}_0$ initialized by $\bar{\boldsymbol{\theta}}_0 \sim \tilde{\rho}_0 = \rho_0$. We prove that (5) is closely approximated by (4) with small step size in the sense that $\mathbb{D}_{\text{KL}}[\tilde{\rho}_k \parallel \bar{\rho}_{k\eta}] \to 0$ as $\eta \to 0$ in Theorem 4.2, and importantly, the stationary distribution of (5) equals to the target distribution $\rho^*(\boldsymbol{\theta}) \propto \exp(-V(\boldsymbol{\theta}))$.

## 4.2 ASSUMPTIONS

We first introduce the techinical assumptions used in our theoretical analysis.

**Assumption 4.1.** *(RBF Kernel)*

*We use RBF kernel i.e.* $K(\boldsymbol{\theta}_1, \boldsymbol{\theta}_2) = \exp(-\|\boldsymbol{\theta}_1 - \boldsymbol{\theta}_2\|^2 / \sigma)$ *for some fixed* $0 < \sigma < \infty$.

We only assume the RBF kernel for the simplicity of our analysis. However, it is straightforward to generalize our theory to other positive definite kernels.

**Assumption 4.2.** *(V is dissipative and smooth)*

*Assume that* $\langle \boldsymbol{\theta}, -\nabla V(\boldsymbol{\theta}) \rangle \leq b_1 - a_1 \|\boldsymbol{\theta}\|^2$ *and* $\|\nabla V(\boldsymbol{\theta}_1) - \nabla V(\boldsymbol{\theta}_2)\| \leq b_1 \|\boldsymbol{\theta}_1 - \boldsymbol{\theta}_2\|$. *We also assume that* $\|\nabla V(\mathbf{0})\| \leq b_1$. *Here* $a_1$ *and* $b_1$ *are some finite positive constant.*

**Assumption 4.3.** *(Regularity Condition)*

*Assume* $\mathbb{E}_{\boldsymbol{\theta} \sim \rho_0}[\|\boldsymbol{\theta}\|^2] < 0$. *Define* $\rho_k^M = \sum_{j=1}^M \rho_{k-jc_\eta}/M$, *assume there exists* $B < \infty$ *such that*

$$\inf_{k \geq Mc_\eta} \left( \sup_{\|\boldsymbol{\theta}\| \leq B} \mathbb{E} \left\| \boldsymbol{g}(\boldsymbol{\theta}; \tilde{\delta}_k^M) - \boldsymbol{g}(\boldsymbol{\theta}; \rho_k^M) \right\|^2 \right) > 0.$$

**Assumption 4.4.** *(Strong-convexity)*

*Suppose that* $\langle \nabla V(\boldsymbol{\theta}_1) - \nabla V(\boldsymbol{\theta}_2), \boldsymbol{\theta}_1 - \boldsymbol{\theta}_2 \rangle \geq L \|\boldsymbol{\theta}_1 - \boldsymbol{\theta}_2\|^2$ *for a positive constant L.*

**Remark**   Assumption 4.2 is standard in the existing Langevin dynamics analysis (see Dalalyan (2017); Raginsky et al. (2017) for example). Assumption 4.3 is a weak condition as it assumes that the dynamics can not degenerate into one local mode and stop moving anymore. This assumption is expected to be true when we have diffusion terms like the Gaussian noises in our self-repulsive dynamics. Assumption 4.4 is a classical assumption on the existing Langevin dynamics analysis with convex potential Dalalyan (2017); Durmus et al. (2019). Although being a bit strong, this assumption broadly applies to posterior inference problem in the limit of big data, as the posterior distribution converges to Gaussian distributions for large training set as shown by Bernstein-von Mises theorem. It is technically possible to further generalize our results to the non-convex settings with a refined analysis, which we leave as future work. This work focuses on the classic convex setting for simplicity.

### 4.3 MAIN THEOREMS

All of the proof in this section can be found in Appendix B.5.

We first prove that the limiting distribution of the continuous-time mean field dynamics (5) is the target distribution. This is achieved by writing dynamics (5) into the following (non-linear) partial differential equation:

$$\partial_t \bar{\rho}_t = \begin{cases} \nabla \cdot (-\nabla V \bar{\rho}_t) + \Delta \bar{\rho}_t & t \in [0, Mc) \\ \nabla \cdot \left[ \left( -\nabla V + \alpha \boldsymbol{g}(\cdot, \bar{\rho}_t^M) \right) \bar{\rho}_t \right] + \Delta \bar{\rho}_t, & t \geq Mc. \end{cases}$$

**Theorem 4.1.** *(Stationary Distribution)*

*Given some finite $M$, $c$ and $\alpha$, and suppose that the limiting distribution of dynamics (5) exists. Then the limit distribution is unique, and equals to $\rho^*(\boldsymbol{\theta}) \propto \exp(-V(\boldsymbol{\theta}))$.*

We then give the upper bound on the discretization error, which can be characterized by analyzing the KL divergence between $\tilde{\rho}_k$ and $\bar{\rho}_{k\eta}$.

**Theorem 4.2.** *(Time Discretization Error)*

*Given some sufficiently small step size $\eta$ and choose $\alpha < a_2/(2b_1 + 4/\sigma)$. Under assumption 4.1, 4.2, 4.3 and $c_\eta = c/\eta$. we have for some constant $C$,*

$$\max_{l \in \{0, \ldots, k\}} \mathbb{D}_{\mathrm{KL}} \left[ \bar{\rho}_{l\eta} \parallel \tilde{\rho}_l \right] \leq \begin{cases} \mathcal{O} \left( \eta + k\eta^2 \right) & k \leq Mc_\eta - 1 \\ \mathcal{O} \left( \eta + Mc\eta + \alpha^2 Mc e^{C\alpha^2 (k\eta - Mc)} \eta^2 \right) & k \geq Mc_\eta. \end{cases}$$

With this theorem, we can know that if $\eta$ is small enough, then the discretization error is small and $\tilde{\rho}$ approximates $\bar{\rho}$ closely. Next we give result on the mean field limit of $M \to \infty$.

**Theorem 4.3.** *(Mean-Field Limit)*

*Under assumption 4.1, 4.2, 4.3, and 4.4, suppose that we choose $\alpha$ and $\eta$ such that $-(a_1 - 2\alpha b_1/\sigma) + \eta b_1 < 0$; $\frac{2\alpha\eta}{\sigma}(b_1 + 1) < 1$; $a_2 - \alpha \left( 2b_1 + \frac{4}{\sigma} \right) > 0$; Then there exists a constant $c_2$, such that when $L/a \geq c_2$ and we have*

$$\mathbb{W}_2^2[\rho_k, \tilde{\rho}_k] = \mathcal{O} \left( \alpha^2/M + \eta^2 \right) \quad for \quad k \geq Mc_\eta,$$

*and $\mathbb{W}_2^2[\rho_k, \tilde{\rho}_k] = 0$ if $k \leq Mc_\eta - 1$.*

**Remark**  Combine all the result, we have $\lim_{k, M \to \infty, \eta \to 0^+} \mathbb{D}_{\mathrm{BL}} \left[ \rho_k, \rho^* \right] = 0$, where the joint limit requires $k\eta \to \infty$, $\exp(C\alpha^2 k\eta)\eta^2 = o(1)$ and $\frac{k\eta}{Mc} = \gamma (1 + o(1))$, with $\gamma > 1$. Notice that if $\gamma \leq 1$, the dynamics is degenerated to Langevin dynamics.

## 5 EXTENSION TO GENERAL DYNAMICS

Although we have focused on self-repulsive Langevin dynamics, our Stein self-repulsive idea can be broadly combined with general gradient-based MCMC. Following Ma et al. (2015), we consider the following general class of sampling dynamics for drawing samples from $p(\boldsymbol{\theta}) \propto \exp(-V(\boldsymbol{\theta}))$:

$$d\boldsymbol{\theta}_t = -\boldsymbol{f}(\boldsymbol{\theta})dt + \sqrt{2\boldsymbol{D}(\boldsymbol{\theta})}d\mathcal{B}_t,$$

with $\boldsymbol{f}(\boldsymbol{\theta}) = [\boldsymbol{D}(\boldsymbol{\theta}) + \boldsymbol{Q}(\boldsymbol{\theta})]\nabla V(\boldsymbol{\theta}) - \boldsymbol{\Gamma}(\boldsymbol{\theta})$,      $\Gamma_i(\boldsymbol{\theta}) = \sum_{j=1}^{d} \frac{\partial}{\partial \boldsymbol{\theta}_j} \left( D_{ij}(\boldsymbol{\theta}) + Q_{ij}(\boldsymbol{\theta}) \right).$

where $\boldsymbol{D}$ is a positive semi-definite diffusion matrix that determines the strength of the Brownian motion and $\boldsymbol{Q}$ is a skew-symmetric curl matrix that can represent the traverse effect (e.g. in Neal et al., 2011; Ding et al., 2014). By adding the Stein repulsive force, we obtain the following general self-repulsive dynamics

$$d\bar{\boldsymbol{\theta}}_t = \begin{cases} -\boldsymbol{f}(\boldsymbol{\theta})dt + \sqrt{2\boldsymbol{D}(\boldsymbol{\theta})}d\mathcal{B}_t, & t \in [0, Mc) \\ - \left( \boldsymbol{f}(\boldsymbol{\theta}) + \alpha \boldsymbol{g}(\bar{\boldsymbol{\theta}}_t; \bar{\rho}_t^M) \right) dt + d\mathcal{B}_t, & t \geq Mc \end{cases} \tag{6}$$

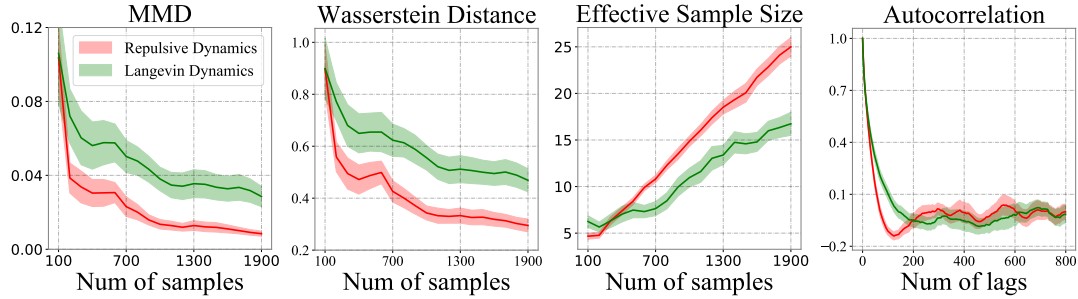

Figure 3: Sample quality of SRLD and Langevin dynamics for sampling the correlated 2D distribution. The auto-correlation is the averaged auto-correlation of the two dimensions.

where $\bar{\rho}_t := \mathrm{Law}(\bar{\boldsymbol{\theta}}_t)$ is again the distribution of $\bar{\boldsymbol{\theta}}_t$ following (6) when initalized at $\bar{\boldsymbol{\theta}}_0 \sim \bar{\rho}_0$. Similar to the case of Langevin dynamics, this process also converges to the correct target distribution, and can be simulated by practical dynamics similar to (3).

**Theorem 5.1.** *(Stationary Distribution)*

*Given some finite $M$, $c$ and $\alpha$, and suppose that the limiting distribution of dynamics (6) exists. Then the limiting distribution is unique and equals the target distribution $\rho^*(\boldsymbol{\theta}) \propto \exp(-V(\boldsymbol{\theta}))$.*

## 6 EXPERIMENTS

In this section, we evaluate the proposed method in various challenging problems, including sampling the posteriors of Bayesian Neural Networks, and uncertainty estimation in Reinforcement Learning. Our results show that our Self-Repulsive Langevin dynamics (SRLD) yields much higher sample efficiency than vanilla Langevin dynamics. Our code is available along the submission.

### 6.1 SYNTHETIC EXPERIMENT

**A Correlated 2D Distribution** This experiment aims to show how the repulsive gradient helps explore the whole distribution. Following Ma et al. (2015), we compare the sampling efficiency on the following correlated 2D distribution with density

$$\rho^*([\theta_1, \theta_2]) \propto -\theta_1^4/10 - \left(4\left(\theta_2 + 1.2\right) - \theta_1^2\right)^2/2.$$

We compare the SRLD with vanilla Langevin dynamics, and evaluate the sample quality by Maximum Mean Discrepancy (MMD) (Gretton et al., 2012), Wasserstein-1 Distance and effective sample size (ESS). Notice that the finite sample quality of gradient based MCMC method is highly related to the step size. Compared with Langevin dynamics, we have an extra repulsive gradient and thus we implicitly have larger step size. To rule out this effect, we set different step sizes of the two dynamics so that the gradient of the two dynamics has the same magnitude. In addition, to decrease the influence of random noise, the two dynamics are set to have the same initialization and use the same sequence of Gaussian noise. We collect the sample of every iteration. We repeat the experiment 20 times with different initialization and sequence of Gaussian noise.

Figure 3 summarizes the result with different metrics. We can see that SRLD have a significantly smaller MMD and Wasserstein-1 Distance as well as a larger ESS compared with the vanilla Langevin dynamics. Moreover, the introduced repulsive gradient creates a negative auto-correlation between samples. Figure 4 shows a typical trajectory of the two sampling dynamics. We can see that SRLD have a faster mixing rate than vanilla Langevin dynamics. Note that since we use the same sequence of Gaussian noise for both algorithms, the difference is mainly due to the use of repulsive gradient rather than the randomness.

**Mixture of Gaussian Distribution** We aim to show how the repulsive gradient helps the particle escape from the local high density region by sampling the 2D mixture of Gaussian distribution using SRLD and Langevin dynamics. The target density is set to be

$$\rho^*(\boldsymbol{\theta}) \propto \frac{1}{2} \exp\left(-\|\boldsymbol{\theta} - \mathbf{1}\|^2/2\right) + \frac{1}{2} \exp\left(-\|\boldsymbol{\theta} + \mathbf{1}\|^2/2\right), \quad \boldsymbol{\theta} = [\theta_1, \theta_2]^\top,$$

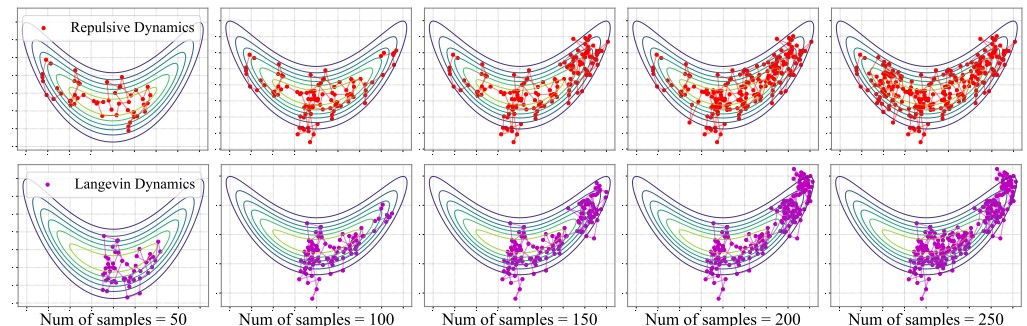

Figure 4: Sampling trajectory of the correlated 2D distribution.

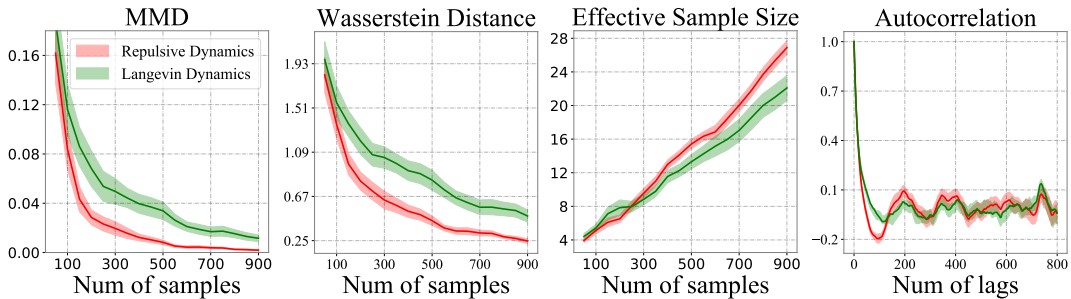

Figure 5: Sample quality and autocorrelation of the mixture distribution. The auto-correlation is the averaged auto-correlation of the two dimensions.

where $\mathbf{1} = [1, 1]^{\top}$. This target distribution have two mode at $-\mathbf{1}$ and $\mathbf{1}$, and vanilla Langevin dynamics can stuck in one mode while keeps the another mode under-explored (as the gradient of energy function can dominate the update of samples). We use the same evaluation method, step sizes, initialization and Gaussian noise as the previous experiment. We collect one sample every 100 iterations and the experiment is repeated for 20 times. Figure 5 shows that SRLD again consistently outperforms the Langevin dynamics on all of the evaluation metrics.

## 6.2 BAYESIAN NEURAL NETWORK

Bayesian Neural Network is one of the most important methods in Bayesian Deep Learning with wide application in practice. Here we test the performance of SRLD on sampling the posterior of Bayesian Neural Network on the UCI datasets (Dua & Graff, 2017). We assume the output is normal distributed, with a two-layer neural network with 50 hidden units and $\tanh$ activation to predict the mean of outputs. We set a $\Gamma(1, 0.1)$ prior for the inverse output variance. All of the datasets are randomly partitioned into $90\%$ for training and $10\%$ for testing. The results are averaged over 20 random trials. We set the mini-batch size to be 100 and the number of past samples $M$ to be 10. In all experiments, we use RBF kernel with bandwidth set by the median trick as suggested in Liu & Wang (2016). For SVGD, we use the original implementation with 20 particles by Liu & Wang (2016). We run 50000 iterations for each methods, and for LD and SRLD, the first 40000 iteration is discarded as burn-in. We use a thinning factor of $c_{\eta} = c/\eta = 100$ and in total we collect 100 samples from the posterior distribution. For each dataset, we generate 3 extra data splits for tuning the step size for each method.

Table 1 shows the average test RMSE and test log-likelihood and their standard deviation. The method that has the best average performance is marked as boldface. We observe that a large portion of the variance is due to the random partition of the dataset. Therefore, to show the statistical significance, we use the matched pair $t$-test to test the statistical significance, mark the methods that perform the best with a significance level of 0.05 with underlines. Note that the results of SRLD/LD and SVGD is not very comparable, because SRLD/LD are single chain methods which averages across time, and SVGD is a multi-chain method that only use the results of the last iteration. We

| Dataset | Ave Test RMSE | | | Ave Test LL | | |
|---|---|---|---|---|---|---|
| | SVGD | LD | SRLD | SVGD | LD | SRLD |
| Boston | $3.300 \pm 0.142$ | $3.342 \pm 0.187$ | $\mathbf{3.086 \pm 0.181}$ | $-4.276 \pm 0.217$ | $-2.678 \pm 0.092$ | $\underline{\mathbf{-2.500 \pm 0.054}}$ |
| Concrete | $4.994 \pm 0.171$ | $4.908 \pm 0.113$ | $\mathbf{4.886 \pm 0.108}$ | $-5.500 \pm 0.398$ | $-3.055 \pm 0.035$ | $\underline{\mathbf{-3.034 \pm 0.031}}$ |
| Energy | $0.428 \pm 0.016$ | $0.412 \pm 0.016$ | $\mathbf{0.395 \pm 0.016}$ | $-0.781 \pm 0.094$ | $-0.543 \pm 0.014$ | $\underline{\mathbf{-0.476 \pm 0.036}}$ |
| Naval | $0.006 \pm 0.000$ | $0.006 \pm 0.002$ | $\mathbf{0.003 \pm 0.000}$ | $3.056 \pm 0.034$ | $4.041 \pm 0.030$ | $\underline{\mathbf{4.186 \pm 0.015}}$ |
| WineRed | $0.655 \pm 0.008$ | $0.649 \pm 0.009$ | $\mathbf{0.639 \pm 0.009}$ | $-1.040 \pm 0.018$ | $-1.004 \pm 0.019$ | $\underline{\mathbf{-0.970 \pm 0.016}}$ |
| WineWhite | $\mathbf{0.655 \pm 0.008}$ | $0.692 \pm 0.003$ | $0.688 \pm 0.003$ | $\underline{\mathbf{-1.040 \pm 0.019}}$ | $-1.047 \pm 0.004$ | $-1.043 \pm 0.004$ |
| Yacht | $0.593 \pm 0.071$ | $0.597 \pm 0.051$ | $\mathbf{0.578 \pm 0.054}$ | $-1.281 \pm 0.279$ | $-1.187 \pm 0.307$ | $\underline{\mathbf{-0.458 \pm 0.036}}$ |

Table 1: Averaged test RMSE and test log-likelihood on UCI datasets. Results are averaged over 20 trails. The boldface indicates the method has the best average performance and the underline marks the methods that perform the best with a significance level of $0.05$.

provide additional results in appendix that SRLD averaged on 20 particles (across time) can also achieve similar or better results as SVGD with 20 (parallel) particles.

## 6.3 Contextual Bandits

We evaluate the quality of uncertainty estimation provided by our methods on several contextual bandits problems. Uncertainty estimation is a key component of contextual bandits. If the agent makes decisions with a poorly estimated uncertainty, the decisions will finally lead to catastrophic failure through the feedback loops (Riquelme et al., 2018).

Though in principle all of the MCMC methods return the samples follow the true posterior if we can run infinite MCMC steps, in practice we can only obtain finite samples as we only have finite time to run the MCMC sampler. In this case, the auto-correlation issue can lead to the under-estimate the uncertainty, which will cause the failure on all of the reinforcement learning problems that need exploration.

We consider the posterior sampling (a.k.a Thompson sampling) algorithm with Bayesian neural network as the function approximator. We follows the experimental setting from Riquelme et al. (2018). The only difference is that we change the optimization of the objective (e.g. evidence lower bound (ELBO) in variational inference methods) into running MCMC samplers. We set the step of samplers equal to the optimization step, and use a thinning factor of 100. We compare the SRLD with the Langevin dynamics on the Mushroom and Wheel bandits from (Riquelme et al., 2018), and include SVGD as a baseline. For more detailed introduction and setup, see Appendix A.5.

The cumulative regret is shown in Table 2. SVGD is known to have the under-estimated uncertainty for Bayesian neural network if particle number is limited (Wang et al., 2019b), and as a result, has the worst performance among the three methods. SRLD is slightly better than vanilla Langevin dynamics on the simple Mushroom bandits. On the much more harder Wheel bandits, SRLD is significantly better than the vanilla Langevin dynamics, which shows the improving uncertainty estimation of our methods within finite number of samples.

| Dataset | SVGD | LD | SRLD |
|---|---|---|---|
| Mushroom | $20.7 \pm 2.0$ | $4.28 \pm 0.09$ | $\mathbf{3.80 \pm 0.16}$ |
| Wheel | $91.32 \pm 0.17$ | $38.07 \pm 1.11$ | $\underline{\mathbf{32.08 \pm 0.75}}$ |

Table 2: Cumulative Regrets on two bandits problem. Results are averaged over 10 trails. Boldface indicates the methods with best performance and underline marks the best significant methods with significant level $0.05$.

## 7 Conclusion

We propose a Stein self-repulsive dynamics which applies Stein variational gradient to push samples from MCMC dynamics away from its past trajectories. This allows us to significantly decrease the auto-correlation of MCMC, increasing the sample efficiency for better estimation. The advantages of our method are extensive studied both theoretical and empirical analysis in our work. In future work, we plan to investigate the combination of our Stein self-repulsive idea with more general MCMC procedures, and explore broader applications.

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

# A   EXPERIMENT DETAILS AND ADDITIONAL EXPERIMENT RESULT

## A.1   SYNTHETIC 2D MIXTURE OF GAUSSIAN EXPERIMENT

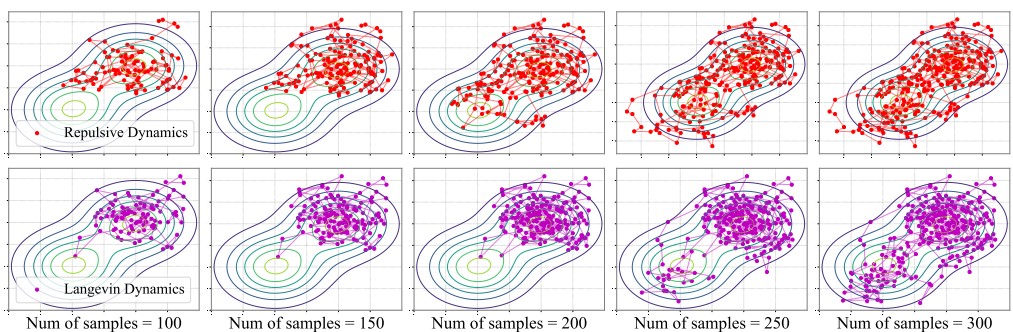

Figure 6: Sampling trajectory of the mixture of Gaussian.

To provide more evidence on the effectiveness of SRLD on escaping from local high density region, we plot the sampling trajectory of SRLD and vanilla Langevin dynamics on the mixture of Gaussian mentioned in Section 6.1. We can find that, when both of the methods obtain 200 samples, SRLD have started to explore the second mode, while vanilla Langevin dynamics still stuck in the original mode. When both of the methods have 250 examples, the vanilla Langevin dynamics just start to explore the second mode, while our SRLD have already obtained several samples from the second mode, which shows our methods effectiveness on escaping the local mode.

## A.2   SYNTHETIC HIGHER DIMENSIONAL GAUSSIAN EXPERIMENT

To show the performance of SRLD in higher dimensional case with different value of $\alpha$, we additionally considering the problem on sampling from Gaussian distribution with $d = 100$ and covariance $\mathbf{\Sigma} = 0.5\mathbf{I}$. We run SRLD with $\alpha = 100, 50, 20, 10, 0$ and the case $\alpha = 0$ reduces to Langevin. We collect 1 sample every 10 iterations. The other experiment setting is the same as the toy examples in the main text. The results are summarized at Figure [7]. In this experiment, we set one SRLD with an inappropriate $\alpha = 100$. For this chain, the repulsive gradient gives strong repulsive force and thus has the largest ESS and the fastest decay of autocorrelation. While the inappropriate value $\alpha$ induces too much extra approximation error and thus its performance is not as good as these with smaller $\alpha$ (see MMD and Wasserstein distance). This phenomenon matches our theoretical finding.

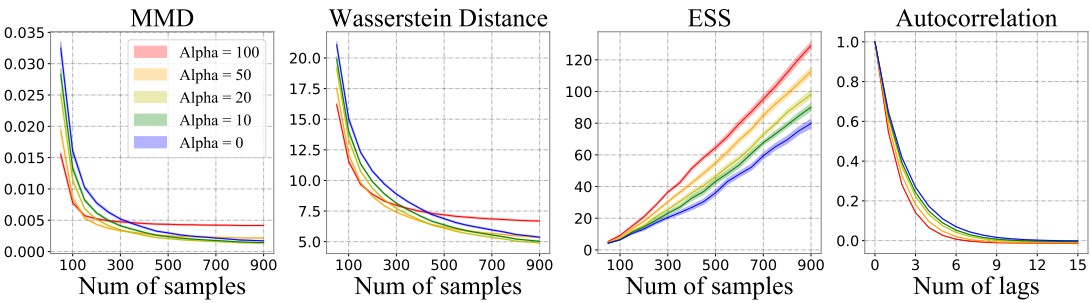

Figure 7: Sample quality and autocorrelation of the higher dimensional Gaussian distribution. The auto-correlation is the averaged auto-correlation of all dimensions.

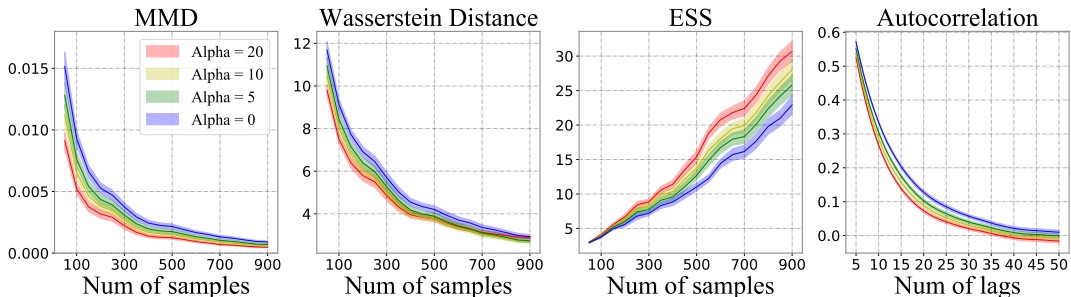

Figure 8: Sample quality and autocorrelation of the higher dimensional mixture distribution. The auto-correlation is the averaged auto-correlation of all dimensions.

### A.3 SYNTHETIC HIGHER DIMENSIONAL MIXTURE OF GAUSSIAN EXPERIMENT

We also consider sampling from the mixture of Gaussian with $d = 20$. The target density is set to be

$$\rho^*(\boldsymbol{\theta}) \propto \frac{1}{2} \exp\left(-0.5 \left\|\boldsymbol{\theta} - \sqrt{2/d}\mathbf{1}\right\|^2\right) + \frac{1}{2} \exp\left(-0.5 \left\|\boldsymbol{\theta} + \sqrt{2/d}\mathbf{1}\right\|^2\right),$$

where $\boldsymbol{\theta} = [\theta_1, ..., \theta_{20}]^\top$ and $\mathbf{1} = [1, ..., 1]^\top$. And thus the mean of the two mixture component is with distance $2\sqrt{2}$. We run SRLD with $\alpha = 20, 10, 5, 0$ (and when $\alpha = 0$, it reduces to LD). The other experiment setting is the same as the low dimensional mixture Gaussian case. Figure [8] summarizes the result. As shown in the figure, when $\alpha$ becomes larger, the repulsive forces helps the sampler better explore the density region.

### A.4 UCI DATASETS

We show some additional experiment result on posterior inference on UCI datasets. As mentioned in Section 6.2, the comparison between SVGD and SRLD is not direct as SVGD is a multiple-chain method with fewer particles and SRLD is a single chain method with more samples. To show more detailed comparison, we compare the SVGD with SRLD using the first 20, 40, 60, 80 and 100 samples, denoted as SRLD-$n$ where $n$ is the number of samples used. Table 3 shows the result of averaged test RMSE and table 4 shows the result of averaged test loglikelihood. For SRLD with different number of samples, the value is set to be boldface if it has better average performance than SVGD. If it is statistical significant with significant level 0.05 using a matched pair t-test, we add an underline on it.

Figure 9 and 10 give some visualized result on the comparison with Langevin dynamics and SRLD. To rule out the variance of different splitting on the dataset, the errorbar is calculated based on the difference between RMSE of SRLD and RMSE of Langevin dynamcis in 20 repeats (And similarily for test log-likelihood). And we only applied the error bar on Langevin dynamics.

| Dataset | Ave Test RMSE | | | | | |
|---|---|---|---|---|---|---|
| | SRLD-20 | SRLD-40 | SRLD-60 | SRLD-80 | SRLD-100 | SVGD |
| Boston | $\mathbf{3.236 \pm 0.174}$ | $\mathbf{3.173 \pm 0.176}$ | $\mathbf{3.130 \pm 0.173}$ | $\mathbf{3.101 \pm 0.179}$ | $\mathbf{3.086 \pm 0.181}$ | $3.300 \pm 0.142$ |
| Concrete | $\mathbf{4.959 \pm 0.109}$ | $\mathbf{4.921 \pm 0.111}$ | $\mathbf{4.906 \pm 0.109}$ | $\mathbf{4.891 \pm 0.108}$ | $\mathbf{4.886 \pm 0.108}$ | $4.994 \pm 0.171$ |
| Energy | $\mathbf{0.422 \pm 0.016}$ | $\mathbf{0.409 \pm 0.016}$ | $\mathbf{0.405 \pm 0.016}$ | $\mathbf{0.399 \pm 0.016}$ | $\mathbf{0.395 \pm 0.016}$ | $0.428 \pm 0.016$ |
| Naval | $\mathbf{0.005 \pm 0.001}$ | $\underline{\mathbf{0.004 \pm 0.000}}$ | $\underline{\mathbf{0.003 \pm 0.000}}$ | $\underline{\mathbf{0.003 \pm 0.000}}$ | $\underline{\mathbf{0.003 \pm 0.000}}$ | $0.006 \pm 0.000$ |
| WineRed | $\mathbf{0.654 \pm 0.009}$ | $\underline{\mathbf{0.647 \pm 0.009}}$ | $\underline{\mathbf{0.644 \pm 0.009}}$ | $\underline{\mathbf{0.641 \pm 0.009}}$ | $\underline{\mathbf{0.639 \pm 0.009}}$ | $0.655 \pm 0.008$ |
| WineWhite | $0.695 \pm 0.003$ | $0.692 \pm 0.003$ | $0.690 \pm 0.003$ | $0.689 \pm 0.002$ | $0.688 \pm 0.003$ | $\mathbf{0.655 \pm 0.008}$ |
| Yacht | $0.616 \pm 0.055$ | $0.608 \pm 0.052$ | $0.597 \pm 0.051$ | $\mathbf{0.587 \pm 0.054}$ | $\mathbf{0.578 \pm 0.054}$ | $0.593 \pm 0.071$ |

Table 3: Comparing SRLD with different number of samples with SVGD on test RMSE. The results are computed over 20 trials. For SRLD, the value is set to be boldface if it has better average performance than SVGD. The value if with underline if it is significantly better than SVGD with significant level 0.05 using a matched pair t-test.

| Dataset | Ave Test LL | | | | | |
| | SRLD-20 | SRLD-40 | SRLD-60 | SRLD-80 | SRLD-100 | SVGD |
|---|---|---|---|---|---|---|
| Boston | $-2.642 \pm .088$ | $-2.582 \pm 0.084$ | $-2.527 \pm 0.612$ | $-2.516 \pm 0.062$ | $-2.500 \pm 0.054$ | $-4.276 \pm 0.217$ |
| Concrete | $-3.084 \pm 0.036$ | $-3.061 \pm 0.034$ | $-3.050 \pm 0.033$ | $-3.040 \pm 0.031$ | $-3.034 \pm 0.031$ | $-5.500 \pm 0.398$ |
| Energy | $-0.580 \pm 0.053$ | $-0.536 \pm 0.048$ | $-0.522 \pm 0.046$ | $-0.504 \pm 0.044$ | $-0.476 \pm 0.036$ | $-0.781 \pm 0.094$ |
| Naval | $4.033 \pm 0.230$ | $4.100 \pm 0.171$ | $4.140 \pm 0.015$ | $4.167 \pm 0.014$ | $4.186 \pm 0.015$ | $3.056 \pm 0.034$ |
| WineRed | $-1.008 \pm 0.019$ | $-0.990 \pm 0.017$ | $-0.982 \pm 0.016$ | $-0.974 \pm 0.016$ | $-0.970 \pm 0.016$ | $-1.040 \pm 0.018$ |
| WineWhite | $-1.053 \pm 0.004$ | $-1.049 \pm 0.004$ | $-1.047 \pm 0.004$ | $-1.044 \pm 0.004$ | $-1.043 \pm 0.004$ | $-1.040 \pm 0.019$ |
| Yacht | $-1.160 \pm 0.256$ | $-0.650 \pm 0.173$ | $-0.556 \pm 0.096$ | $-0.465 \pm 0.037$ | $-0.458 \pm 0.036$ | $-1.281 \pm 0.279$ |

Table 4: Comparing SRLD with different number of samples with SVGD on test log-likelihood. The results are computed over 20 trials. For SRLD, the value is set to be boldface if it has better average performance than SVGD. The value if with underline if it is significantly better than SVGD with significant level 0.05 using a matched pair t-test.

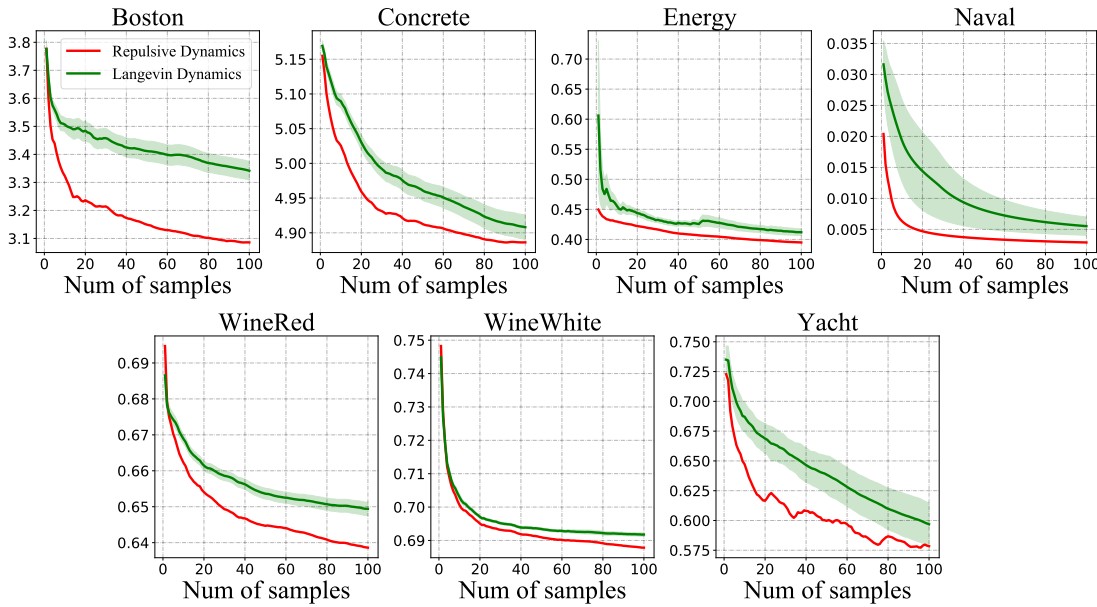

Figure 9: Comparison between SRLD and Langevin dynamics on test RMSE. The results are computed based on 20 repeats. The error bar is calculated based on RMSE of SRLD - RMSE of Langevin dynamics in 20 repeats to rule out the variance of different data splitting

## A.5 CONTEXTUAL BANDIT

Contextual bandit is a class of online learning problems that can be viewed as a simple reinforcement learning problem without transition. For a completely understanding of contextual bandit problems, we refer the readers to the Chapter 4 of (Bubeck et al., 2012). Here we include the main idea for completeness. In contextual bandit problems, the agent needs to find out the best action given some observed context (a.k.a the optimal policy in reinforcement learning). Formally, we define $\mathcal{S}$ as the context set and $K$ as the number of action. Then we can concretely describe the contextual bandit problems as follows: for each time-step $t = 1, 2, \cdots, N$, where $N$ is some pre-defined time horizon (and can be given to the agent), the environment provides a context $s_t \in \mathcal{S}$ to the agent, then the agent should choose one action $a_t \in \{1, 2, \cdots, K\}$ based on context $s_t$. The environment will return a (stochastic) reward $r(s_t, a_t)$ to the agent based on the context $s_t$ and the action $a_t$ that similar to the reinforcement learning setting. And notice that, the agent can adjust the strategy at each time-step, so that this kinds of problems are called "online" learning problem.

Solving the contextual bandit problems is equivalent to find some algorithms that can minimize the pseudo-regret (Bubeck et al., 2012), which is defined as:

$$\overline{R}_N^{\mathcal{S}} = \max_{\pi:\mathcal{S} \rightarrow \{1,2,\cdots,K\}} \mathbb{E}\left[\sum_{t=1}^N r(s_t, g(s_t)) - \sum_{t=1}^N r(s_t, a_t)\right]. \tag{7}$$

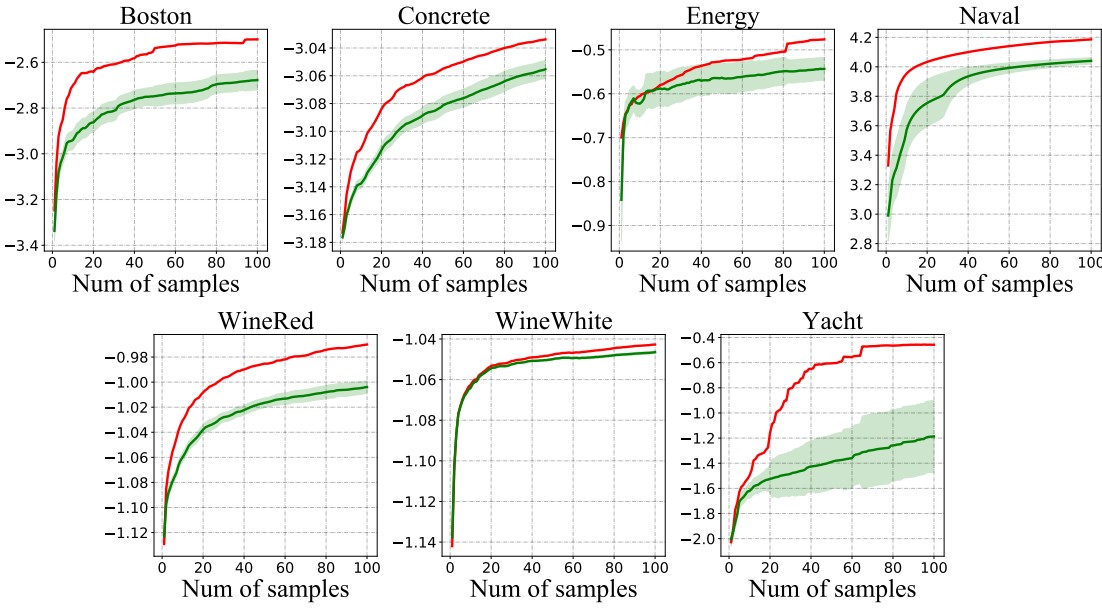

Figure 10: Comparison between SRLD and Langevin dynamics on test log-likelihood. The results are computed based on 20 repeats. The error bar is calculated based on log-likelihood of SRLD - log-likelihood of Langevin dynamics in 20 repeats to rule out the variance of data splitting.

where $\pi$ denotes the deterministic mapping from the context set $\mathcal{S}$ to actions $\{1, 2, \cdots, K\}$ (readers can view $\pi$ as a deterministic policy in reinforcement learning). Intuitively, this pseudo-regret measures the difference of cumulative reward between the action sequence $a_t$ and the best action sequence $\pi(s_t)$. Thus, an algorithm that can minimize the pseudo-regret (7) can also find the best $\pi$.

Posterior sampling (a.k.a. Thompson sampling; Thompson, 1933) is one of the classical yet successful algorithms that can achieve the state-of-the-art performance in practice (Chapelle & Li, 2011). It works by first placing an user-specified prior $\mu_{s,a}^0$ on the reward $r(s, a)$, and each turn make decision based on the posterior distribution and update it, i.e. update the posterior distribution $\mu_{s,a}^t$ with the observation $r(s_{t-1}, a_{t-1})$ at time $t-1$ where $a_{t-1}$ is selected with the posterior distribution: each time, the action is selected with the following way:

$$a_t = \underset{a \in \{1,2,\cdots,K\}}{\arg\max} \hat{r}(s_t, a), \quad \hat{r}(s_t, a) \sim \mu_{s,a}^t.$$

i.e., greedy select the action based on the sampled reward from the posterior, thus called "Posterior Sampling". Algorithm 1 summarizes the whole procedure of Posterior Sampling.

---

**Algorithm 1** Posterior sampling for contextual bandits

---

**Input:** Prior distribution $\mu_{s,a}^0$, time horizon $N$
**for** time $t = 1, 2, \cdots, N$ **do**
    observe a new context $s_t \in \mathcal{S}$,
    sample the reward of each action $\hat{r}(s_t, a) \sim \mu_{s,a}^t, a \in \{1, 2, \cdots, K\}$,
    select action $a_t = \arg\max_{a \in \{1,2,\cdots,K\}} \hat{r}(s_t, a)$ and get the reward $r(s_t, a_t)$,
    update the posterior $\mu_{s_t,a_t}^{t+1}$ with $r(s_t, a_t)$.
**end for**

---

Notice that all of the reinforcement learning problems face the *exploration-exploitation dilemma*, so as the contextual bandit problem. Posterior sampling trade off the exploration and exploitation with the uncertainty provided by the posterior distribution. So if the posterior uncertainty is not estimated properly, posterior sampling will perform poorly. To see this, if we over-estimate the uncertainty, we can explore too-much sub-optimal actions, while if we under-estimate the uncertainty, we can

fail to find the optimal actions. Thus, it is a good benchmark for evaluating the uncertainty provided by different inference methods.

Here, we test the uncertainty provided by vanilla Langevin dynamics and Self-repulsive Langevin dynamics on two of the benchmark contextual bandit problems suggested by (Riquelme et al., 2018), called *mushroom* and *wheel*. One can read (Riquelme et al., 2018) to find the detail introduction of this two contextual bandit problems. For completeness, we include it as follows:

**Mushroom** Mushroom bandit utilizes the data from Mushroom dataset (Schlimmer, 1981), which includes different kinds of poisonous mushroom and safe mushroom with 22 attributes that can indicate whether the mushroom is poisonous or not. Blundell et al. (2015) first introduced the mushroom bandit by designing the following reward function: eating a safe mushroom will give a $+5$ reward, while eating a poisonous mushroom will return a reward $+5$ and $-35$ with equal chances. The agent can also choose not to eat the mushroom, which always yield a $0$ reward. Same to (Riquelme et al., 2018), we use 50000 instances in this problem.

**Wheel** To highlight the need for exploration, (Riquelme et al., 2018) designs the wheel bandit, that can control the need of exploration with some "exploration parameter" $\delta \in (0, 1)$. The context set $\mathcal{S}$ is the unit circle $\|s\|_2 \leq 1$ in $\mathbb{R}^2$, and each turn the context $s_t$ is uniformly sampled from $\mathcal{S}$. $K = 5$ possible actions are provided: the first action yields a constant reward $r \sim \mathcal{N}(\mu_1, \sigma^2)$; the reward corresponding to other actions is determined by the provided context $s$:

- For $s \in \mathcal{S}$ s.t. $\|s\|_2 \leq \delta$, all of the four other actions return a suboptimal reward sampled from $\mathcal{N}(\mu_2, \sigma^2)$ for $\mu_2 < \mu_1$.
- For $s \in \mathcal{S}$ s.t. $\|s\|_2 > \delta$, according to the quarter the context $s$ is in, one of the four actions becomes optimal. This optimal action gives a reward of $\mathcal{N}(\mu_3, \sigma^2)$ for $\mu_3 \gg \mu_1$, and another three actions still yield the suboptimal reward $\mathcal{N}(\mu_2, \sigma^2)$.

Following the setting from (Riquelme et al., 2018), we set $\mu_1 = 1.2$, $\mu_2 = 1.0$, and $\mu_3 = 50$.

When $\delta$ approaches 1, the inner circle $\|s\|_2 \leq \delta$ will dominate the unit circle and the first action becomes the optimal for most of the context. Thus, inference methods with poorly estimated uncertainty will continuously choose the suboptimal action $a_1$ for all of the contexts without exploration. This phenomenon have been confirmed in (Riquelme et al., 2018). In our experiments, as we want to evaluate the quality of uncertainty provided by different methods, we set $\delta = 0.95$, which is pretty hard for existing inference methods as shown in (Riquelme et al., 2018), and use 50000 contexts for evaluation.

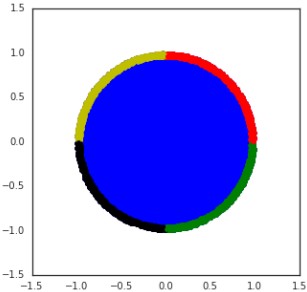

Figure 11: Visualization of the wheel bandit ($\delta = 0.95$), taken from (Riquelme et al., 2018).

**Experiment Setup** Following (Riquelme et al., 2018), we use a feed-forward network with two hidden layer of 100 units and ReLU activation. We use the same step-size and thinning factor $c/\eta = 100$ for vanilla Langevin dynamics and SRLD, and set $M = 20$, $\alpha = 1$ on both of the mushroom and wheel bandits. The update schedule is similar to (Riquelme et al., 2018), and we just change the optimization step in stochastic variational inference methods into MCMC sampler step and use a higher (2000) initial steps for burn-in like the warm-up of stochastic variational inference methods in Riquelme et al. (2018). As this is an online posterior inference problem, we only use the

last 20 samples to give the prediction. Notice that, in the original implementation of Riquelme et al. (2018), the authors only update a few steps with new observation after observing enough data, as the posterior will gradually converge to the true reward distribution and little update is needed after observing sufficient data. Similar to their implementation, after observing enough data, we only collect one new sample with the new observation each time. For SVGD, we also use 20 particles to make the comparison fair.

## B  THE DETAILED ANALYSIS OF SRLD

### B.1  SOME ADDITIONAL NOTATION

We use $\| \cdot \|_\infty$ to denote the $\ell_\infty$ vector norm and define the $\mathcal{L}_\infty$ norm of a function $f : \mathbb{R}^d \to \mathbb{R}^1$ as $\|f\|_{\mathcal{L}_\infty}$. $\mathbb{D}_{\mathrm{TV}}$ denote the Total Variation distance between distribution $\rho_1, \rho_2$ respectively. Also, as $K$ is $\mathbb{R}^d \times \mathbb{R}^d \to \mathbb{R}^1$, we denote $\|K\|_{\mathcal{L}_\infty, \mathcal{L}_\infty} = \sup_{\boldsymbol{\theta}_1, \boldsymbol{\theta}_2} K(\boldsymbol{\theta}_1, \boldsymbol{\theta}_2)$. For simplicity, we may use $\|K\|_{\infty, \infty}$ as $\|K\|_{\mathcal{L}_\infty, \mathcal{L}_\infty}$. In the appendix, we also use $\phi[\rho](\boldsymbol{\theta}) := \boldsymbol{g}(\boldsymbol{\theta}; \rho)$, where $\boldsymbol{g}(\boldsymbol{\theta}; \rho)$ is defined in the main text. For the clearance, we define $\pi_{M, c/\eta} * \rho_k := \rho_k^M$, $\pi_{M, c/\eta} * \tilde{\rho}_k := \tilde{\rho}_k^M$ and $\pi_{M, c} * \bar{\rho}_t := \bar{\rho}_t^M$, where $\rho_k^M, \tilde{\rho}_k^M$ and $\bar{\rho}_t^M$ are defined in main text.

### B.2  GEOMETRIC ERGODICITY OF SRLD

Before we start the proof of main theorems, we give the following theorem on the geometric ergodicity of SRLD. It is noticeable that under this assumption, the practical dynamics follows an $(Mc/\eta + 1)$-order nonlinear autoregressive model when $k \geq Mc/\eta$:

$$\boldsymbol{\theta}_{k+1} = \boldsymbol{\psi}\left(\boldsymbol{\theta}_k, ..., \boldsymbol{\theta}_{k-Mc/\eta}\right) + \sqrt{2\eta}\boldsymbol{e}_k,$$

where

$$\boldsymbol{\psi}\left(\boldsymbol{\theta}_k, ..., \boldsymbol{\theta}_{k-Mc/\eta}\right) = \boldsymbol{\theta}_k + \eta \left\{ -\nabla V(\boldsymbol{\theta}_k) + \alpha\phi[\frac{1}{M} \sum_{j=1}^{M} \delta_{\boldsymbol{\theta}_{k-jc/\eta}}](\boldsymbol{\theta}_k) \right\}.$$

Further, if we stack the parameter by $\boldsymbol{\Theta}_k = \left[\boldsymbol{\theta}_k, ..., \boldsymbol{\theta}_{k-Mc/\eta}\right]^\top$ and define $\boldsymbol{\Psi}\left(\boldsymbol{\Theta}_k\right) = \left[\boldsymbol{\psi}^\top\left(\boldsymbol{\Theta}_k\right), \boldsymbol{\Theta}_k^\top\right]^\top$, we have

$$\boldsymbol{\Theta}_{k+1} = \boldsymbol{\Psi}\left(\boldsymbol{\Theta}_k\right) + \sqrt{2\eta}\boldsymbol{E}_k,$$

where $\boldsymbol{E}_k = \left[\boldsymbol{e}_k^\top, \boldsymbol{0}^\top, ..., \boldsymbol{0}^\top\right]^\top$. In this way, we formulate $\boldsymbol{\Theta}_k$ as a time homogeneous Markov Chain. In the following analysis, we only analyze the second phase of SRLD given some initial stacked particles $\boldsymbol{\Theta}_{Mc/\eta-1}$.

**Theorem B.1.** *(Geometric Ergodicity) Under Assumption 4.1 and Assumption 4.2, suppose we choose $\eta$ and $\alpha$ such that*

$$\max\left(1 - 2\eta a_1 + \eta^2 b_1 + \frac{2\alpha\eta}{\sigma} b_1, \frac{2\alpha\eta}{\sigma}(b_1 + 1)\right) < 1,$$

*then the Markov Chain of $\boldsymbol{\Theta}_k$ is stationary, geometrically ergodic, i.e., for any $\boldsymbol{\Theta}_0' = \boldsymbol{\Theta}_{Mc/\eta-1}$, we have*

$$\mathbb{D}_{\mathrm{TV}}\left[P^k\left(\cdot, \boldsymbol{\Theta}_0\right), \Pi\left(\cdot\right)\right] \leq Q\left(\boldsymbol{\Theta}_0\right)e^{-rk},$$

*where $r$ is some positive constant, $Q(\boldsymbol{\Theta}_0)$ is constant related to $\boldsymbol{\Theta}_0$, $P^k$ is the $k$-step Markov transition kernel and $\Pi$ is the stationary distribution.*

We defer the proof to Appendix B.5.1.

### B.3  MOMENT BOUND

**Theorem B.2.** *(Moment Bound) Under assumption 4.2, suppose that we have $\mathbb{E}_{\boldsymbol{\theta} \sim \rho_0} \|\boldsymbol{\theta}\|^2 < \infty$; and $a_2 - \alpha \|K\|_\infty \left(2b_1 + \frac{4}{\sigma}\right) > 0$, we have*

$$\sup_k \mathbb{E}_{\boldsymbol{\theta} \sim \rho_k} \|\boldsymbol{\theta}\|^2 \vee \sup_k \mathbb{E}_{\boldsymbol{\theta} \sim \tilde{\rho}_k} \|\boldsymbol{\theta}\|^2 \vee \sup_t \mathbb{E}_{\boldsymbol{\theta} \sim \bar{\rho}_t} \|\boldsymbol{\theta}\|^2$$

$$\leq \mathbb{E}_{\boldsymbol{\theta} \sim \rho_0} \|\boldsymbol{\theta}\|^2 + \frac{b_1 + 1 + \eta}{a_2 - \|K\|_{\mathcal{L}_\infty, \mathcal{L}_\infty} \frac{2\alpha}{\sigma} - \alpha \|K\|_{\mathcal{L}_\infty, \mathcal{L}_\infty}\left(2b_1 + \frac{2}{\sigma}\right)}.$$

*And by Lemma B.1, we thus have*

$$\sup_k \mathbb{E}_{\boldsymbol{\theta} \sim \rho_k} \|\nabla V(\boldsymbol{\theta})\|^2 \vee \sup_k \mathbb{E}_{\boldsymbol{\theta} \sim \tilde{\rho}_k} \|\nabla V(\boldsymbol{\theta})\|^2 \vee \sup_t \mathbb{E}_{\boldsymbol{\theta} \sim \bar{\rho}_t} \|\nabla V(\boldsymbol{\theta})\|^2$$

$$\leq b_1 \mathbb{E}_{\boldsymbol{\theta} \sim \rho_0} \|\boldsymbol{\theta}\|^2 + \frac{b_1(b_1 + 1 + \eta)}{a_2 - \|K\|_{\mathcal{L}_\infty, \mathcal{L}_\infty} \frac{2\alpha}{\sigma} - \alpha \|K\|_{\mathcal{L}_\infty, \mathcal{L}_\infty}\left(2b_1 + \frac{2}{\sigma}\right)} + 1$$

The proof can be found at Appendix B.5.2.

## B.4 TECHNICAL LEMMA

**Definition B.1.** *($\alpha$-mixing)*

*For any two $\sigma$-algebras $\mathcal{A}$ and $\mathcal{B}$, the $\alpha$-mixing coefficient is defined by*
$$\alpha(\mathcal{A}, \mathcal{B}) = \sup_{A \in \mathcal{A}, B \in \mathcal{B}} |\mathbb{P}(A \cap B) - \mathbb{P}(A)\mathbb{P}(B)|.$$

*Let $(X_k, k \geq 1)$ be a sequence of real random variable defined on $(\Omega, \mathcal{A}, \mathbb{P})$. This sequence is $\alpha$-mixing if*
$$\alpha(n) := \sup_{k \geq 1} \alpha(\mathcal{M}_k, \mathcal{G}_{k+n}) \to 0, \text{ as } n \to \infty,$$

*where $\mathcal{M}_j := \sigma(X_i, i \leq j)$ and $\mathcal{G}_j := \sigma(X_i, i \geq j)$ for $j \geq 1$. Alternatively, as shown by Theorem 4.4 of Bradley (2007)*
$$\alpha(n) := \frac{1}{4} \sup \left\{ \frac{\text{Cov}(f, g)}{\|f\|_{\mathcal{L}_\infty} \|g\|_{\mathcal{L}_\infty}}; \ f \in \mathcal{L}_\infty(\mathcal{M}_k), \ g \in \mathcal{L}_\infty(\mathcal{G}_{k+n}) \right\}.$$

**Definition B.2.** *($\beta$-mixing)*

*For any two $\sigma$-algebras $\mathcal{A}$ and $\mathcal{B}$, the $\alpha$-mixing coefficient is defined by*
$$\beta(\mathcal{A}, \mathcal{B}) := \sup \frac{1}{2} \sum_{i=1}^{I} \sum_{j=1}^{J} |\mathbb{P}(A_i \cap B_j) - \mathbb{P}(A_i)\mathbb{P}(B_j)|,$$

*where the supremum is taken over all pairs of finite partitions $\{A_1, ..., A_I\}$ and $\{B_1, ..., B_J\}$ of $\Omega$ such that $A_i \in \mathcal{A}$ and $B_j \in \mathcal{B}$ for each $i$, $j$. Let $(X_k, k \geq 1)$ be a sequence of real random variable defined on $(\Omega, \mathcal{A}, \mathbb{P})$. This sequence is $\beta$-mixing if*
$$\beta(n) := \sup_{k \geq 1} \beta(\mathcal{M}_k, \mathcal{G}_{k+n}) \to 0, \text{ as } n \to \infty.$$

**Proposition B.1.** *($\beta$-mixing implies $\alpha$-mixing)*

*For any two $\sigma$-algebras $\mathcal{A}$ and $\mathcal{B}$,*
$$\alpha(\mathcal{A}, \mathcal{B}) \leq \frac{1}{2}\beta(\mathcal{A}, \mathcal{B}).$$

This proposition can be found in Equation 1.11 of Bradley (2005).

**Proposition B.2.** *A (strictly) stationary Markov Chain is geometric ergodicity if and only if $\beta(n) \to 0$ at least exponentially fast as $n \to \infty$.*

This proposition is Theorem 3.7 of Bradley (2005).

**Lemma B.1.** *By Assumption 4.2, we have $\|\nabla V(\boldsymbol{\theta})\| \leq b_1(\|\boldsymbol{\theta}_1\| + 1)$ and $\|\boldsymbol{\theta} - \eta \nabla V(\boldsymbol{\theta})\| \leq (1 - 2\eta a_1 + \eta^2 b_1) \|\boldsymbol{\theta}\|^2 + \eta^2 b_1 + 2\eta b_1$.*

**Lemma B.2.** *(Some property of RBF Kernel) For RBF kernel with bandwidth $\sigma$, we have $\|K\|_{\infty,\infty} \leq 1$ and*
$$\|K(\boldsymbol{\theta}', \boldsymbol{\theta}_1) - K(\boldsymbol{\theta}', \boldsymbol{\theta}_2)\| \leq \left\| e^{-(\cdot)^2/\sigma} \right\|_{\text{Lip}} \|\boldsymbol{\theta}_1 - \boldsymbol{\theta}_2\|_2$$

$$\|\nabla_{\boldsymbol{\theta}'} K(\boldsymbol{\theta}', \boldsymbol{\theta}_1) - \nabla_{\boldsymbol{\theta}'} K(\boldsymbol{\theta}', \boldsymbol{\theta}_2)\| \leq \left\| \frac{2}{\sigma} e^{-(\cdot)^2/\sigma}(\cdot) \right\|_{\text{Lip}} \|\boldsymbol{\theta}_1 - \boldsymbol{\theta}_2\|_2.$$

**Lemma B.3.** *(Some property of Stein Operator)*

*For any distribution $\rho$ such that $\mathbb{E}_{\boldsymbol{\theta} \sim \rho} \|\nabla V(\boldsymbol{\theta})\| < \infty$, we have*
$$\|\phi[\rho](\cdot)\|_{\text{Lip}} \leq \left\| e^{-(\cdot)^2/\sigma} \right\|_{\text{Lip}} \mathbb{E}_{\boldsymbol{\theta} \sim \rho} \|\nabla V(\boldsymbol{\theta})\| + \left\| \frac{2}{\sigma} e^{-(\cdot)^2/\sigma}(\cdot) \right\|_{\text{Lip}},$$

$$\|\phi[\rho](\boldsymbol{\theta})\| \leq \|K\|_\infty \mathbb{E}_{\boldsymbol{\theta}' \sim \rho} \left[ \|\nabla V(\boldsymbol{\theta}')\| + \frac{2}{\sigma}(\|\boldsymbol{\theta}'\| + \|\boldsymbol{\theta}\|) \right]$$

$$\leq \|K\|_\infty b_1 + \mathbb{E}_{\boldsymbol{\theta}' \sim \rho} \left[ \left( \frac{2}{\sigma} + b_1 \right) \|\boldsymbol{\theta}'\| \right] + \|\boldsymbol{\theta}\|.$$

**Lemma B.4.** *(Bounded Lipschitz of Stein Operator) Given $\theta'$, define $\bar{\phi}_{\theta'}(\theta) := \phi[\delta_{\theta'}](\theta) = K(\theta', \theta)\nabla V(\theta') + \nabla_1 K(\theta', \theta)$. We also denote $\bar{\phi}_{\theta'}(\theta) = [\bar{\phi}_{\theta',1}(\theta), ..., \bar{\phi}_{\theta',d}(\theta)]^\top$. We have*

$$\sum_{i=1}^{d} \left\| \bar{\phi}_{\theta',i}(\theta) \right\|_{\mathrm{Lip}}^2 \leq 2 \left\| \nabla V(\theta') \right\|^2 \left\| e^{-\|\cdot\|^2/\sigma} \right\|_{\mathrm{Lip}}^2 + 2d \left\| \frac{2}{\sigma} e^{-\|\theta\|^2/\sigma} \theta_1 \right\|_{\mathrm{Lip}}^2$$

$$\sum_{i=d}^{d} \left\| \bar{\phi}_{\theta',i}(\theta) \right\|_{\mathcal{L}_\infty}^2 \leq 2d \left\| \frac{2}{\sigma} e^{-\|\theta\|^2/\sigma} \theta_1 \right\|_{\mathcal{L}_\infty}^2 + 2 \left\| e^{-\|\cdot\|^2/\sigma} \right\|_{\mathcal{L}_\infty}^2 \left\| \nabla V(\theta') \right\|^2 .$$

### B.5 PROOF OF MAIN THEOREMS

#### B.5.1 PROOF OF THEOREM B.1

The proof of this theorem is by verifying the condition of Theorem 3.2 of An & Huang (1996). Suppose $\Theta = [\theta_1, ..., \theta_{MC+1}]$, where $C = c/\eta$, we have

$$\|\psi(\Theta)\| = \left\| \theta_1 + \eta \left\{ -\nabla V(\theta_1) + \alpha \phi[\frac{1}{M} \sum_{j=1}^{M} \delta_{\theta_{1+jC}}](\theta_k) \right\} \right\|$$

$$= \left\| \theta_1 - \eta \nabla V(\theta_1) + \frac{\eta \alpha}{M} \sum_{j=1}^{M} \left[ e^{-\|\theta_{1+jC} - \theta_1\|^2/\sigma} \frac{2}{\sigma} (\theta_1 - \theta_{1+jC}) - e^{-\|\theta_{1+jC} - \theta_1\|^2/\sigma} \nabla V(\theta_{1+jC}) \right] \right\|$$

$$\leq \left\| \theta_1 - \eta \nabla V(\theta_1) + \frac{2}{\sigma} \frac{\eta \alpha}{M} \sum_{j=1}^{M} e^{-\|\theta_{1+jC} - \theta_1\|^2/\sigma} \theta_1 \right\|$$

$$+ \left\| \frac{\eta \alpha}{M} \sum_{j=1}^{M} e^{-\|\theta_{1+jC} - \theta_1\|^2/\sigma} \frac{2}{\sigma} (-\nabla V(\theta_{1+jC}) - \theta_{1+jC}) \right\|$$

$$\leq \|\theta_1 - \eta \nabla V(\theta_1)\| + \frac{2\alpha\eta}{\sigma} \|K\|_{\infty,\infty} b_1 (1 + \|\theta_1\|)$$

$$+ \frac{2\alpha\eta}{M\sigma} \sum_{j=1}^{M} \|K\|_{\infty,\infty} b_1 \left( 1 + (1 + \frac{1}{b_1}) \|\theta_{1+jC}\| \right)$$

$$\overset{(1)}{\leq} b_1 (1 + \frac{4\alpha\eta}{\sigma} \|K\|_{\infty,\infty}) + \eta^2 b_1 + 2\eta b_1$$

$$+ \left( 1 - 2\eta a_1 + \eta^2 b_1 + \frac{2\alpha\eta}{\sigma} \|K\|_{\infty,\infty} b_1 \right) \|\theta_1\| + \frac{2\alpha\eta}{\sigma} \|K\|_{\infty,\infty} (b_1 + 1) \max_{i \in [MC+1]-\{1\}} \|\theta_{1+jC}\|$$

$$\leq b_1 (1 + \frac{4\alpha\eta}{\sigma} \|K\|_{\infty,\infty}) + \eta^2 b_1 + 2\eta b_1$$

$$+ \max \left( 1 - 2\eta a_1 + \eta^2 b_1 + \frac{2\alpha\eta}{\sigma} \|K\|_{\infty,\infty} b_1, \frac{2\alpha\eta}{\sigma} \|K\|_{\infty,\infty} (b_1 + 1) \right) \max_{i \in [MC+1]} \|\theta_{1+jC}\|,$$

where (1) is by Lemma B.1. Thus, given the step size $\eta$, if we choose $\eta, \alpha$ such that

$$\max \left( 1 - 2\eta a_1 + \eta^2 b_1 + \frac{2\alpha\eta}{\sigma} \|K\|_{\infty,\infty} b_1, \frac{2\alpha\eta}{\sigma} \|K\|_{\infty,\infty} (b_1 + 1) \right) < 1,$$

then our dynamics is geometric ergodic.

#### B.5.2 PROOF OF THEOREM B.2

**Continuous-Time Mean Field Dynamics** (5) Notice that as our dynamics has two phases and the first phase can be viewed as an special case of the second phase by setting $\alpha = 0$, here we only analysis the second phase. Define $U_t = \sup_{s \leq t} \sqrt{\mathbb{E} \|\bar{\theta}_s\|^2}$, and thus

$$\frac{\partial}{\partial t} U_t^2 \leq \mathbb{E} \left\langle \bar{\theta}_t, -V(\bar{\theta}) + \alpha \phi[\pi_{M,c} * \bar{\rho}_t](\bar{\theta}_t) \right\rangle \vee 0.$$

Now we bound $\mathbb{E}\left\langle \bar{\boldsymbol{\theta}}_t, -V(\bar{\boldsymbol{\theta}}) + \alpha\phi[\pi_{M,c} * \bar{\rho}_t](\bar{\boldsymbol{\theta}}_t)\right\rangle$:

$$
\begin{aligned}
&\mathbb{E}\left\langle \bar{\boldsymbol{\theta}}_t, -V(\bar{\boldsymbol{\theta}}_t) + \alpha\phi[\pi_{M,c} * \bar{\rho}_t](\bar{\boldsymbol{\theta}}_t)\right\rangle \\
&\leq b_1 - a_2\mathbb{E}\left\|\bar{\boldsymbol{\theta}}_t\right\|^2 + \alpha\mathbb{E}\left\|\bar{\boldsymbol{\theta}}_t\right\|\left\|\phi[\pi_{M,c} * \bar{\rho}_t](\bar{\boldsymbol{\theta}}_t)\right\| \\
&\overset{(1)}{\leq} b_1 - a_2\mathbb{E}\left\|\bar{\boldsymbol{\theta}}_t\right\|^2 + \alpha\left\|K\right\|_\infty \mathbb{E}\left\|\bar{\boldsymbol{\theta}}_t\right\| \mathbb{E}_{\boldsymbol{\theta}'\sim\pi_{M,c}*\bar{\rho}_t}\left[\left\|\nabla V(\boldsymbol{\theta}')\right\| + \frac{2}{\sigma}\left(\left\|\boldsymbol{\theta}'\right\| + \left\|\boldsymbol{\theta}_t\right\|\right)\right] \\
&\leq b_1 - a_2\mathbb{E}\left\|\bar{\boldsymbol{\theta}}_t\right\|^2 + \alpha\left\|K\right\|_\infty \mathbb{E}\left\|\bar{\boldsymbol{\theta}}_t\right\| \mathbb{E}_{\boldsymbol{\theta}'\sim\pi_{M,c}*\bar{\rho}_t}\left[b_1\left(\left\|\boldsymbol{\theta}'\right\| + 1\right) + \frac{2}{\sigma}\left(\left\|\boldsymbol{\theta}'\right\| + \left\|\boldsymbol{\theta}_t\right\|\right)\right] \\
&= b_1 - \left(a_2 - \left\|K\right\|_\infty \frac{2\alpha}{\sigma}\right)\mathbb{E}\left\|\bar{\boldsymbol{\theta}}_t\right\|^2 + \alpha\left\|K\right\|_\infty \mathbb{E}\left\|\bar{\boldsymbol{\theta}}_t\right\| \mathbb{E}_{\boldsymbol{\theta}'\sim\pi_{M,c}*\bar{\rho}_t}\left(\left(b_1 + \frac{2}{\sigma}\right)\left\|\boldsymbol{\theta}'\right\| + b_1\right) \\
&\leq b_1 - \left(a_2 - \left\|K\right\|_\infty \frac{2\alpha}{\sigma}\right)U_t^2 + \alpha\left\|K\right\|_\infty \mathbb{E}\left\|\bar{\boldsymbol{\theta}}_t\right\| \mathbb{E}_{\boldsymbol{\theta}'\sim\pi_{M,c}*\bar{\rho}_t}\left(\left(b_1 + \frac{2}{\sigma}\right)\left\|\boldsymbol{\theta}'\right\| + b_1\right) \\
&\leq b_1 - \left(a_2 - \left\|K\right\|_\infty \frac{2\alpha}{\sigma}\right)U_t^2 + \alpha\left\|K\right\|_\infty\left(b_1 + \frac{2}{\sigma}\right)\frac{1}{M}\sum_{j=1}^M U_t U_{t-jc} + \alpha\left\|K\right\|_\infty b_1 U_t \\
&\leq b_1 - \left(a_2 - \left\|K\right\|_\infty \frac{2\alpha}{\sigma}\right)U_t^2 + \alpha\left\|K\right\|_\infty\left(b_1 + \frac{2}{\sigma}\right)U_t^2 + \alpha\left\|K\right\|_\infty b_1(U_t^2 + 1) \\
&\leq (b_1 + 1) - \left(a_2 - \left\|K\right\|_\infty \frac{2\alpha}{\sigma} - \alpha\left\|K\right\|_\infty\left(2b_1 + \frac{2}{\sigma}\right)\right)U_t^2,
\end{aligned}
$$

where (1) is by B.3. By the assumption that $\lambda := a_2 - \left\|K\right\|_\infty \frac{2\alpha}{\sigma} - \alpha\left\|K\right\|_\infty\left(2b_1 + \frac{2}{\sigma}\right) > 0$, we have

$$
\frac{\partial}{\partial t}U_t^2 \leq \left[(b_1 + 1) - \lambda U_t^2\right] \vee 0.
$$

By Gronwall's inequality, we have $U_t^2 \leq U_0^2 + \frac{b_1+1}{\lambda}$. (If $\frac{\partial}{\partial t}U_t^2 = 0$, then $U_t$ fix and this bound still holds.) Notice that in the first phase, as $\alpha = 0$, we have $\lambda < a_2$ and thus this inequality also holds.

**Discrete-Time Mean Field Dynamics (4)** Similarly to the analysis of the continuous-time mean field dynamics (5), we only give proof of the second phase. Define $U_k = \sup_{s\leq k}\sqrt{\mathbb{E}\left\|\tilde{\boldsymbol{\theta}}_s\right\|^2}$, and thus

$$
U_k^2 - U_{k-1}^2 \leq \left[2\eta\mathbb{E}\left\langle \tilde{\boldsymbol{\theta}}_{k-1}, -\nabla V(\tilde{\boldsymbol{\theta}}_k) + \alpha\phi[\pi_{M,c/\eta} * \tilde{\rho}_k](\tilde{\boldsymbol{\theta}}_k)\right\rangle + 2\eta^2\right] \vee 0.
$$

By a similarly analysis, we have bound

$$
\begin{aligned}
&\mathbb{E}\left\langle \tilde{\boldsymbol{\theta}}_{k-1}, -\nabla V(\tilde{\boldsymbol{\theta}}_k) + \alpha\phi[\pi_{M,c/\eta} * \tilde{\rho}_k](\tilde{\boldsymbol{\theta}}_k)\right\rangle \\
&\leq (b_1 + 1) - \lambda U_t^2,
\end{aligned}
$$

where $\lambda = a_2 - \left\|K\right\|_{\infty,\infty}\frac{2\alpha}{\sigma} - \alpha\left\|K\right\|_{\infty,\infty}\left(2b_1 + \frac{2}{\sigma}\right) > 0$. And thus we have

$$
U_k^2 - U_{k-1}^2 \leq \left[2\eta\left[(b_1 + 1) - \lambda U_{k-1}^2\right] + 2\eta^2\right] \vee 0.
$$

It gives that

$$
U_k^2 \leq \frac{b_1 + 1 + \eta}{\lambda} + U_0^2.
$$

**Practical Dynamics (3)** The analysis of Practical Dynamics (3) is almost identical to that of the discrete-time mean field dynamics (4) and thus is omitted here.

### B.5.3  PROOF OF THEOREM 4.1 AND 5.1

Notice that the dynamics in Theorem 4.1 is special case of that in Theorem 5.1 and thus we only prove Theorem 5.1 here. After some algebra, we can show that the continuity equation of dynamics (6) is

$$
\partial_t\rho_t = \nabla\cdot\left(\left[-\left(D(\boldsymbol{\theta}) + Q(\boldsymbol{\theta})\right)\nabla V(\boldsymbol{\theta}) + \alpha\phi[\pi_{M,c} * \rho_t](\boldsymbol{\theta}_t)\right]\rho_t + \left(D(\boldsymbol{\theta}) + Q(\boldsymbol{\theta})\right)\nabla\rho_t\right).
$$

Notice that the limiting distribution satisfies

$$
\begin{aligned}
0 \stackrel{a.e.}{=} & \nabla \cdot \left( \left[ -\left( D(\boldsymbol{\theta}) + Q(\boldsymbol{\theta}) \right) \nabla V(\boldsymbol{\theta}) + \alpha \phi[\pi_{M,c} * \rho_\infty](\boldsymbol{\theta}_t) \right] \rho_\infty + \left( D(\boldsymbol{\theta}) + Q(\boldsymbol{\theta}) \right) \nabla \rho_\infty \right) \\
= & \nabla \cdot \left( \left[ -\left( D(\boldsymbol{\theta}) + Q(\boldsymbol{\theta}) \right) \nabla V(\boldsymbol{\theta}) + \alpha \phi[\rho_\infty](\boldsymbol{\theta}_t) \right] \rho_\infty + \left( D(\boldsymbol{\theta}) + Q(\boldsymbol{\theta}) \right) \nabla \rho_\infty \right) \\
= & \nabla \cdot \left( \left[ -\left( D(\boldsymbol{\theta}) + Q(\boldsymbol{\theta}) \right) \nabla V(\boldsymbol{\theta}) \right] \rho_\infty + \left( D(\boldsymbol{\theta}) + Q(\boldsymbol{\theta}) \right) \nabla \rho_\infty \right) \\
& + \alpha \nabla \cdot \left( K * \left( \nabla \rho_\infty - \nabla V(\boldsymbol{\theta}) \rho_\infty \right) \rho_\infty \right).
\end{aligned}
$$

which implies that $\rho_\infty \propto \exp(-V(\boldsymbol{\theta}))$ is the stationary distribution.

### B.5.4 PROOF OF THEOREM 4.2

In the later proof we use $c_d$ to represent the quantity

$$
\sqrt{ \mathbb{E}_{\boldsymbol{\theta} \sim \rho_0} \| \boldsymbol{\theta} \|^2 + \frac{b_1 + 1 + \eta}{a_2 - \| K \|_{\infty,\infty} \frac{2\alpha}{\sigma} - \alpha \| K \|_{\infty,\infty} \left( 2b_1 + \frac{2}{\sigma} \right)} }.
$$

Recall that there are two dynamics: the continuous-time mean field dynamics (5) and the discretized version discrete-time mean field Dynamics (4). Notice that here we couple the discrete-time mean field dynamics with the continuous-time mean field system using the same initialization. Given any $T = \eta N$, for any $0 \le t \le T$, define $\underline{t} = \lfloor \frac{t}{\eta} \rfloor \eta$. We introduce an another continuous-time interpolation dynamics:

$$
\begin{aligned}
\hat{\boldsymbol{\theta}}_t &= \begin{cases} -\nabla V(\hat{\boldsymbol{\theta}}_{\underline{t}}) + d\mathcal{B}_t, & t \in [0, Mc) \\ -\nabla V(\hat{\boldsymbol{\theta}}_{\underline{t}}) + \alpha \phi[\pi_{M,c} * \hat{\rho}_{\underline{t}}](\hat{\boldsymbol{\theta}}_{\underline{t}}) + d\mathcal{B}_t, & t \ge Mc, \end{cases} \\
\hat{\rho}_t &= \mathrm{Law}(\hat{\boldsymbol{\theta}}_t), \\
\hat{\boldsymbol{\theta}}_0 &= \bar{\boldsymbol{\theta}}_0 \sim \bar{\rho}_0,
\end{aligned}
$$

Notice that here we couples this interpolation dynamics with the same Brownian motion as that of the dynamics of $\bar{\boldsymbol{\theta}}_t$. By the definition of $\hat{\boldsymbol{\theta}}_t$, at any $t_k := k\eta$ for some integrate $k \in [N]$, $\hat{\boldsymbol{\theta}}_{t_k}$ and $\tilde{\boldsymbol{\theta}}_k$ has the same distribution. Define $\bar{\rho}_t^{\boldsymbol{\theta}_0} = \mathrm{Law}(\bar{\boldsymbol{\theta}}_t)$ conditioning on $\bar{\boldsymbol{\theta}}_0 = \boldsymbol{\theta}_0$ and $\hat{\rho}_t^{\boldsymbol{\theta}_0} = \mathrm{Law}(\hat{\boldsymbol{\theta}}_t)$ conditioning on $\hat{\boldsymbol{\theta}}_0 = \boldsymbol{\theta}_0$. Followed by the argument of proving Lemma 2 in Dalalyan (2017), if $k \ge \frac{Mc}{\eta}$, we have

$$
\begin{aligned}
& \mathbb{D}_{\mathrm{KL}} \left[ \bar{\rho}_{t_k}^{\boldsymbol{\theta}_0} \,\|\, \hat{\rho}_{t_k}^{\boldsymbol{\theta}_0} \right] \\
=& \frac{1}{4} \int_0^{t_k} \mathbb{E} \left\| -\nabla V(\hat{\boldsymbol{\theta}}_{\underline{s}}) + \alpha \phi[\pi_{M,c} * \hat{\rho}_{\underline{s}}](\hat{\boldsymbol{\theta}}_{\underline{s}}) + \nabla V(\hat{\boldsymbol{\theta}}_s) - \alpha \phi[\pi_{M,c} * \bar{\rho}_s](\hat{\boldsymbol{\theta}}_s) \right\|^2 ds \\
=& \frac{1}{4} \sum_{j=0}^{k-1} \int_{t_j}^{t_{j+1}} \mathbb{E} \left\| -\nabla V(\hat{\boldsymbol{\theta}}_{t_j}) + \alpha \phi[\pi_{M,c} * \hat{\rho}_{t_j}](\hat{\boldsymbol{\theta}}_{t_j}) + \nabla V(\hat{\boldsymbol{\theta}}_s) - \alpha \phi[\pi_{M,c} * \bar{\rho}_s](\hat{\boldsymbol{\theta}}_s) \right\|^2 ds \\
\le& \frac{3}{4} \sum_{j=0}^{k-1} \int_{t_j}^{t_{j+1}} \mathbb{E} \left\| \nabla V(\hat{\boldsymbol{\theta}}_{t_j}) - \nabla V(\hat{\boldsymbol{\theta}}_s) \right\|^2 ds \\
& + \frac{3\alpha^2}{4} \sum_{j=0}^{k-1} \int_{t_j}^{t_{j+1}} \mathbb{E} \left\| \phi[\pi_{M,c} * \hat{\rho}_{t_j}](\hat{\boldsymbol{\theta}}_{t_j}) - \phi[\pi_{M,c} * \bar{\rho}_s](\hat{\boldsymbol{\theta}}_{t_j}) \right\|^2 ds \\
\le& \frac{3\alpha^2}{4} \sum_{j=0}^{k-1} \int_{t_j}^{t_{j+1}} \mathbb{E} \left\| \phi[\pi_{M,c} * \bar{\rho}_s](\hat{\boldsymbol{\theta}}_{t_j}) - \phi[\pi_{M,c} * \bar{\rho}_s](\hat{\boldsymbol{\theta}}_s) \right\|^2 ds \\
=& I_1 + I_2 + I_3.
\end{aligned}
$$

We bound $I_1$, $I_2$ and $I_3$ separately.

**Bounding $I_1$ and $I_3$**  By the smoothness of $\nabla V$, we have

$$
\left\| \nabla V(\hat{\boldsymbol{\theta}}_{t_j}) - \nabla V(\hat{\boldsymbol{\theta}}_s) \right\|^2 \le b_1^2 \left\| \hat{\boldsymbol{\theta}}_{t_j} - \hat{\boldsymbol{\theta}}_s \right\|^2.
$$

And by Lemma B.3 (Lipschitz of Stein Operator), we know that

$$\|\phi[\pi_{M,c} * \bar{\rho}_s](\boldsymbol{\theta}_1) - \phi[\pi_{M,c} * \bar{\rho}_s](\boldsymbol{\theta}_2)\|$$

$$\leq \left[ \left\| e^{-(\cdot)^2/\sigma} \right\|_{\text{Lip}} \mathbb{E}_{\boldsymbol{\theta} \sim \pi_{M,c} * \bar{\rho}_s} \|\nabla V(\boldsymbol{\theta})\| + \left\| \frac{2}{\sigma} e^{-(\cdot)^2/\sigma}(\cdot) \right\|_{\text{Lip}} \right] \|\boldsymbol{\theta}_1 - \boldsymbol{\theta}_2\|_2 .$$

And by the Assumption 4.2 and that $\bar{\rho}_s$ as finite second moment, we have

$$\|\phi[\pi_{M,c} * \bar{\rho}_s](\boldsymbol{\theta}_1) - \phi[\pi_{M,c} * \bar{\rho}_s](\boldsymbol{\theta}_2)\|$$
$$\leq C c_d \|\boldsymbol{\theta}_1 - \boldsymbol{\theta}_2\|_2 .$$

Combine the two bounds, we have

$$I_1 + I_3 \leq \frac{3 C c_d^2}{4} \sum_{j=0}^{k-1} \int_{t_j}^{t_{j+1}} \mathbb{E} \left\| \hat{\boldsymbol{\theta}}_{t_j} - \hat{\boldsymbol{\theta}}_s \right\|^2 ds.$$

Notice that $\hat{\boldsymbol{\theta}}_t = \hat{\boldsymbol{\theta}}_{\underline{t}} + \left[ -\nabla V(\hat{\boldsymbol{\theta}}_{\underline{t}}) + \alpha \phi[\pi_{M,c} * \hat{\rho}_{\underline{t}}](\hat{\boldsymbol{\theta}}_{\underline{t}}) \right] (t - \underline{t}) + \int_{\underline{t}}^t d\mathcal{B}_s$. By Itô's lemma, it implies that

$$I_1 + I_3 \leq \frac{3 C c_d^2}{4} \sum_{j=0}^{k-1} \int_{t_j}^{t_{j+1}} \mathbb{E} \left\| \hat{\boldsymbol{\theta}}_{t_j} - \hat{\boldsymbol{\theta}}_s \right\|^2 ds$$

$$\leq \frac{3 C c_d^2}{4} \int_{t_j}^{t_{j+1}} \left[ \mathbb{E} \left\| -\nabla V(\hat{\boldsymbol{\theta}}_{\underline{s}}) + \alpha \phi[\pi_{M,c} * \hat{\rho}_{\underline{s}}](\hat{\boldsymbol{\theta}}_{\underline{s}}) \right\|^2 (s - t_j)^2 + 2d(s - t_j) \right] ds$$

$$= C c_d^2 \eta^3 \sum_{j=0}^{k-1} \mathbb{E} \left\| -\nabla V(\hat{\boldsymbol{\theta}}_{t_j}) + \alpha \phi[\pi_{M,c} * \hat{\rho}_{t_j}](\hat{\boldsymbol{\theta}}_{t_j}) \right\|^2 + C c_d^2 dk h^2.$$

By the assumption that $\mathbb{E} \left\| \tilde{\boldsymbol{\theta}}_{t_j} \right\|$ is finite and $\tilde{\boldsymbol{\theta}}_{t_j} \overset{d}{=} \hat{\boldsymbol{\theta}}_{t_j}$, $\mathbb{E} \left\| \hat{\boldsymbol{\theta}}_{t_j} \right\|^2$ is also finite, we have

$$\mathbb{E} \left\| -\nabla V(\hat{\boldsymbol{\theta}}_{\underline{t}}) + \alpha \phi[\pi_{M,c} * \hat{\rho}_{\underline{t}}](\hat{\boldsymbol{\theta}}_{\underline{t}}) \right\|^2$$

$$\leq 2 \mathbb{E} \left\| \nabla V(\hat{\boldsymbol{\theta}}_{\underline{t}}) \right\|^2 + 2 \alpha^2 \mathbb{E} \left\| \phi[\pi_{M,c} * \hat{\rho}_{\underline{t}}](\hat{\boldsymbol{\theta}}_{\underline{t}}) \right\|^2$$

$$\leq 4 b_1^2 + 4 b_1^2 \mathbb{E} \left\| \hat{\boldsymbol{\theta}}_{\underline{t}} \right\|^2 + 2 \alpha^2 \mathbb{E} \left( \left( \frac{2}{\sigma} + b_1 \right) \mathbb{E}_{\boldsymbol{\theta}' \sim \pi_{M,c} * \hat{\rho}_{\underline{t}}} \|\boldsymbol{\theta}'\| + \|\boldsymbol{\theta}\| \right)^2$$

$$\leq c_d^2 C.$$

Thus we conclude that

$$I_1 + I_3 \leq C c_d^2 \left( c_d^2 k \eta^3 + dk \eta^2 \right).$$

**Bounding $I_2$**

$$\mathbb{E} \left\| \phi[\pi_{M,c} * \hat{\rho}_{t_j}](\hat{\boldsymbol{\theta}}_{t_j}) - \phi[\pi_{M,c} * \bar{\rho}_s](\hat{\boldsymbol{\theta}}_{t_j}) \right\|^2$$

$$= \mathbb{E} \left\| \frac{1}{M} \sum_{l=1}^{M} \left[ \phi[\hat{\rho}_{t_j - cl}](\hat{\boldsymbol{\theta}}_{t_j}) - \phi[\bar{\rho}_{s-cl}](\hat{\boldsymbol{\theta}}_{t_j}) \right] \right\|^2$$

$$\leq \frac{1}{M} \sum_{l=1}^{M} \mathbb{E} \left\| \phi[\hat{\rho}_{t_j - cl}](\hat{\boldsymbol{\theta}}_{t_j}) - \phi[\bar{\rho}_{s-cl}](\hat{\boldsymbol{\theta}}_{t_j}) \right\|^2$$

$$= \frac{1}{M} \sum_{l=1}^{M} \mathbb{E} \left\| \mathbb{E}_{\boldsymbol{\theta} \sim \hat{\rho}_{t_j - cl}} \bar{\phi}_{\hat{\boldsymbol{\theta}}_{t_j}}(\boldsymbol{\theta}) - \mathbb{E}_{\boldsymbol{\theta} \sim \bar{\rho}_{s-cl}} \bar{\phi}_{\hat{\boldsymbol{\theta}}_{t_j}}(\boldsymbol{\theta}) \right\|^2$$

$$= \frac{1}{M} \sum_{l=1}^{M} \mathbb{E}_{\hat{\boldsymbol{\theta}}_{t_j}} \sum_{i=1}^{d} \left| \mathbb{E}_{\boldsymbol{\theta} \sim \hat{\rho}_{t_j - cl}} \bar{\phi}_{\hat{\boldsymbol{\theta}}_{t_j}, i}(\boldsymbol{\theta}) - \mathbb{E}_{\boldsymbol{\theta} \sim \bar{\rho}_{s-cl}} \bar{\phi}_{\hat{\boldsymbol{\theta}}_{t_j}, i}(\boldsymbol{\theta}) \right|^2$$

$$\leq \frac{1}{M} \sum_{l=1}^{M} \mathbb{E}_{\hat{\boldsymbol{\theta}}_{t_j}} \sum_{i=1}^{d} \left( \left\| \bar{\phi}_{\hat{\boldsymbol{\theta}}_{t_j}, i}(\cdot) \right\|_{\mathcal{L}_\infty} \vee \left\| \bar{\phi}_{\hat{\boldsymbol{\theta}}_{t_j}, i}(\cdot) \right\|_{\text{Lip}} \right)^2 \mathbb{D}_{\text{BL}}^2 \left[ \hat{\rho}_{t_j - cl}, \bar{\rho}_{s-cl} \right]$$

By Lemma B.4 and the Assumption 4.4 that $V$ is at most quadratic growth and that $\hat{\rho}_{\underline{t}}$ has finite second moment, we have

$$
\begin{aligned}
&\mathbb{E}_{\hat{\boldsymbol{\theta}}_{t_j}} \sum_{i=1}^{d} \left( \left\| \bar{\phi}_{\hat{\boldsymbol{\theta}}_{t_j},i}(\cdot) \right\|_{\mathcal{L}_\infty} \vee \left\| \bar{\phi}_{\hat{\boldsymbol{\theta}}_{t_j},i}(\cdot) \right\|_{\mathrm{Lip}} \right)^2 \\
=&\mathbb{E}_{\hat{\boldsymbol{\theta}}_{t_j}} \sum_{i=1}^{d} \left( \left\| \bar{\phi}_{\hat{\boldsymbol{\theta}}_{t_j},i}(\cdot) \right\|_{\mathcal{L}_\infty}^2 \vee \left\| \bar{\phi}_{\hat{\boldsymbol{\theta}}_{t_j},i}(\cdot) \right\|_{\mathrm{Lip}}^2 \right) \\
\leq& \left[ 4d \left\| \frac{2}{\sigma} e^{-\|\boldsymbol{\theta}\|^2/\sigma} \theta_1 \right\|_{\mathrm{BL}}^2 + 4 \left\| e^{-\|\cdot\|^2/\sigma} \right\|_{\mathrm{BL}}^2 \mathbb{E}_{\hat{\boldsymbol{\theta}}_{t_j}} \left\| \nabla V(\hat{\boldsymbol{\theta}}_{t_j}) \right\|^2 \right] \\
\leq& C(d + c_d^2).
\end{aligned}
$$

Plug in the above estimation, we have

$$
\begin{aligned}
I_2 &= \frac{3\alpha^2}{4} \sum_{j=0}^{k-1} \int_{t_j}^{t_{j+1}} \mathbb{E} \left\| \phi[\pi_{M,c} * \hat{\rho}_{t_j}](\hat{\boldsymbol{\theta}}_{t_j}) - \phi[\pi_{M,c} * \bar{\rho}_s](\hat{\boldsymbol{\theta}}_{t_j}) \right\|^2 ds \\
&\leq \alpha^2 C(d + c_d^2) \sum_{j=0}^{k-1} \int_{t_j}^{t_{j+1}} \frac{1}{M} \sum_{l=1}^{M} \mathbb{D}_{\mathrm{BL}}^2 \left[ \hat{\rho}_{t_j-cl}, \bar{\rho}_{s-cl} \right] ds \\
&\leq \alpha^2 C(d + c_d^2) \sum_{j=0}^{k-1} \frac{1}{M} \sum_{l=1}^{M} \int_{t_j}^{t_{j+1}} \mathbb{D}_{\mathrm{KL}} \left[ \hat{\rho}_{t_j-cl}, \bar{\rho}_{s-cl} \right] ds,
\end{aligned}
$$

where the last inequality is due to the relation that $\mathbb{D}_{\mathrm{BL}}^2 \overset{\text{definition}}{\leq} \mathbb{D}_{\mathrm{TV}}^2 \overset{\text{Pinsker's}}{\leq} \mathbb{D}_{\mathrm{KL}}$.

**Overall Bound**  Combine all the estimation, we have

$$
\begin{aligned}
\mathbb{D}_{\mathrm{KL}} \left[ \bar{\rho}_{t_k}^{\boldsymbol{\theta}_0} \| \hat{\rho}_{t_k}^{\boldsymbol{\theta}_0} \right] &\leq \alpha^2 C(d + c_d^2) \sum_{j=0}^{k-1} \frac{1}{M} \sum_{l=1}^{M} \int_{t_j}^{t_{j+1}} \mathbb{D}_{\mathrm{KL}} \left[ \hat{\rho}_{t_j-cl}, \bar{\rho}_{s-cl} \right] ds + Cc_d^2 \left( c_d^2 k\eta^3 + dk\eta^2 \right) \\
&= \alpha^2 C(d + c_d^2) \sum_{j=0}^{k-1} \frac{1}{M} \sum_{l=1}^{M} \int_0^\eta \mathbb{D}_{\mathrm{KL}} \left[ \hat{\rho}_{t\left(\frac{jn-cl}{\eta}\right)}, \bar{\rho}_{t\left(\frac{jn-cl}{\eta}\right)+s} \right] ds + Cc_d^2 \left( c_d^2 k\eta^3 + dk\eta^2 \right)
\end{aligned}
$$

Similar, if $k \leq \frac{Mc}{\eta} - 1$, we have

$$
\begin{aligned}
&\mathbb{D}_{\mathrm{KL}} \left[ \bar{\rho}_{t_k}^{\boldsymbol{\theta}_0} \| \hat{\rho}_{t_k}^{\boldsymbol{\theta}_0} \right] \\
=&\frac{1}{4} \int_0^{t_k} \mathbb{E} \left\| \nabla V(\hat{\boldsymbol{\theta}}_{\underline{s}}) - \nabla V(\hat{\boldsymbol{\theta}}_s) \right\|^2 ds \\
\leq&\frac{b_1^2}{4} \sum_{j=0}^{k-1} \int_{t_j}^{t_{j+1}} \mathbb{E} \left\| \hat{\boldsymbol{\theta}}_{t_j} - \hat{\boldsymbol{\theta}}_s \right\|^2 ds \\
\leq&\frac{b_1^2 \eta^3}{12} \sum_{j=0}^{k-1} \mathbb{E} \left\| \nabla V(\hat{\boldsymbol{\theta}}_{t_j}) \right\|^2 + \frac{dk b_1^2 \eta^2}{4} \\
\leq&\frac{b_1^2 \eta^3 k c_d^2}{12} + \frac{dk b_1^2 \eta^2}{4}.
\end{aligned}
$$

Define

$$
u_k = \sup_{s \in [t_k, t_{k+1}]} \mathbb{D}_{\mathrm{KL}} \left[ \bar{\rho}_{\underline{s}}^{\boldsymbol{\theta}_0} \| \hat{\rho}_s^{\boldsymbol{\theta}_0} \right],
$$

and $U_k = \max\limits_{l \in \{0,\dots,k\}} u_l$. We conclude that for $k \geq \frac{Mc}{\eta}$, for any $k' \leq k$,

$$u_{k'} \leq \alpha^2 C(d + c_d^2) \sum_{j=0}^{k-1} \frac{1}{M} \sum_{l=1}^{M} \int_0^h \mathbb{D}_{\mathrm{KL}} \left[ \hat{\rho}_{t\left(\frac{j\eta - cl}{\eta}\right)}, \bar{\rho}_{t\left(\frac{j\eta - cl}{\eta}\right)+s} \right] ds + Cc_d^2 \left( c_d^2 k\eta^3 + dk\eta^2 \right)$$

$$\leq \alpha^2 C(d + c_d^2) \sum_{j=0}^{k-1} \frac{1}{M} \sum_{l=1}^{M} \eta u_{\left(\frac{j\eta - cl}{\eta}\right)} + Cc_d^2 \left( c_d^2 k\eta^3 + dk\eta^2 \right)$$

$$\leq \alpha^2 C(d + c_d^2)\eta \sum_{j=0}^{k-1} U_j + Cc_d^2 \left( c_d^2 k\eta^3 + dk\eta^2 \right).$$

For $k < \frac{Mc}{\eta}$, which is a simpler case, we have

$$U_k \leq C \left( \eta^3 k c_d^2 + dk\eta^2 \right) < CMc \left( \eta c_d^2 + d \right) \eta.$$

We bound the case when $k \geq \frac{Mc}{\eta}$,

$$U_k \leq \alpha^2 C(d + c_d^2)\eta \sum_{j=0}^{k-1} U_j + Cc_d^2 \left( c_d^2 k\eta^3 + dk\eta^2 \right).$$

If we take $\eta$ sufficiently small, such that $c_d^2 k\eta^3 \leq dk\eta^2$, we have

$$U_k \leq \alpha^2 C(d + c_d^2)\eta \sum_{j=0}^{k-1} U_j + 2Cc_d^2 dk\eta^2$$

$$\leq \alpha^2 C(d + c_d^2)\eta \sum_{j=0}^{k-1} (U_j + \eta).$$

Define $\eta' = \alpha^2 C(d + c_d^2)\eta$ and we can choose $\eta$ small enough such that $\eta' < 1/2$ and $\eta < 1/2$. Without loss of generality, we also assume $\eta' \geq \eta$ and thus we have

$$U_k \leq \eta' \sum_{j=0}^{k-1} (U_j + \eta').$$

Also we assume $U_k \geq \eta'$, otherwise we conclude that $U_k < \eta'$. We thus have $U_k \leq q \sum_{j=0}^{k-1} U_j$, where $q = 2\eta'$. Suppose that $U_{\frac{Mc}{\eta}-1} = x \leq CMc \left( \eta c_d^2 + d \right) \eta$ and some algebra (which reduces to Pascal's triangle) shows that

$$U_k \leq xq(1 + q)^{k - \frac{Mc}{\eta}}.$$

We conclude that $U_k \leq xq(1 + q)^{k-1}$. Notice that $q = 2\alpha^2 C(d + c_d^2)\eta$. Thus for any $k \geq Mc/\eta$,

$$U_k \leq xq(1 + q)^{k - \frac{Mc}{\eta}}$$

$$= xq(1 + q)^{(k\eta - Mc)/\eta}$$

$$= xq(1 + q)^{2\alpha^2 C(d + c_d^2)(k\eta - Mc)/q}$$

$$\leq x2\alpha^2 C(d + c_d^2)e^{2\alpha^2 C(d + c_d^2)(k\eta - Mc)}\eta$$

$$\leq CMc\alpha^2 \left( \eta c_d^2 + d \right) (d + c_d^2)e^{2\alpha^2 C(d + c_d^2)(k\eta - Mc)}\eta^2,$$

for sufficiently small $\eta$. Combine the above two estimations, we have

$$U_k \leq \begin{cases} C \left( \eta^3 k c_d^2 + dk\eta^2 + \eta \right) & k \leq Mc/\eta - 1 \\ CMc\alpha^2 \left( \eta c_d^2 + d \right) (d + c_d^2)e^{2\alpha^2 C(d + c_d^2)(k\eta - Mc)}\eta^2 + C\eta & k \geq Mc/\eta \end{cases}.$$

Notice that now we have $U_k = \max\limits_{l \in \{0,\dots,k\}} \sup\limits_{s \in [0,\eta]} \mathbb{D}_{\mathrm{KL}} \left[ \bar{\rho}_{l\eta+s}^{\boldsymbol{\theta}_0} \| \tilde{\rho}_{l\eta}^{\boldsymbol{\theta}_0} \right]$, which is a function of $\boldsymbol{\theta}_0$.

We then bound $\bar{U}_k = \max\limits_{l \in \{0,\dots,k\}} \sup\limits_{s \in [0,\eta]} \mathbb{D}_{\mathrm{KL}} \left[ \bar{\rho}_{l\eta+s} \| \tilde{\rho}_{l\eta} \right]$. Notice that the KL divergence has the following variational representation:

$$\mathbb{D}_{\mathrm{KL}}[\rho_1 \| \rho_2] = \sup_f \left[ \mathbb{E}_{\rho_1} f - \mathbb{E}_{\rho_2} e^f \right],$$

where the $f$ is chosen in the set that $\mathbb{E}_{\rho_1} f$ and $\mathbb{E}_{\rho_2} e^f$ exist. And thus we have

$$\mathbb{D}_{\mathrm{KL}}[\bar{\rho}_{l\eta+s} \parallel \tilde{\rho}_{l\eta}] = \sup_f \left[ \mathbb{E}_{\boldsymbol{\theta}_0 \sim \rho_0} \left( \mathbb{E}_{\bar{\rho}_{l\eta+s}^{\boldsymbol{\theta}_0}} f - \mathbb{E}_{\tilde{\rho}_{l\eta}^{\boldsymbol{\theta}_0}} e^f \right) \right]$$

$$\leq \mathbb{E}_{\boldsymbol{\theta}_0 \sim \rho_0} \sup_f \left[ \left( \mathbb{E}_{\bar{\rho}_{l\eta+s}^{\boldsymbol{\theta}_0}} f - \mathbb{E}_{\tilde{\rho}_{l\eta}^{\boldsymbol{\theta}_0}} e^f \right) \right].$$

And thus $\bar{U}_k \leq U_k$. Also the inequality that

$$\bar{U}_k = \max_{l \in \{0,\ldots,k\}} \sup_{s \in [0,\eta]} \mathbb{D}_{\mathrm{KL}} \left[ \bar{\rho}_{l\eta+s} \parallel \tilde{\rho}_{l\eta} \right] \geq \max_{l \in \{0,\ldots,k\}} \mathbb{D}_{\mathrm{KL}} \left[ \bar{\rho}_{l\eta} \parallel \tilde{\rho}_{l\eta} \right]$$

holds naturally by definition. We complete the proof.

### B.5.5 PROOF OF THEOREM 4.3

The constant $h_1$ is defined as

$$h_1 = \left\| \frac{2}{\sigma} e^{-\|\boldsymbol{\theta}\|^2/\sigma} \boldsymbol{\theta}_1 \right\|_{\mathrm{BL}}^2 \vee \left\| e^{-\|\cdot\|^2/\sigma} \right\|_{\mathrm{BL}}^2 \vee \left\| \frac{2}{\sigma} e^{-(\cdot)^2/\sigma}(\cdot) \right\|_{\mathrm{Lip}}$$

Now we start the proof. We couple the process of $\boldsymbol{\theta}_k$ and $\tilde{\boldsymbol{\theta}}_k$ by the same gaussian noise $\boldsymbol{e}_k$ in every iteration and same initialization $\tilde{\boldsymbol{\theta}}_0 = \boldsymbol{\theta}_0$. For $k \leq Mc/\eta - 1$, $\mathbb{E} \left\| \boldsymbol{\theta}_k - \tilde{\boldsymbol{\theta}}_k \right\|^2 = 0$ and for $k \geq Mc/\eta$ we have the following inequality,

$$\mathbb{E} \left\| \boldsymbol{\theta}_{k+1} - \tilde{\boldsymbol{\theta}}_{k+1} \right\|^2 - \mathbb{E} \left\| \boldsymbol{\theta}_k - \tilde{\boldsymbol{\theta}}_k \right\|^2$$

$$= 2\eta \mathbb{E} \left\langle \boldsymbol{\theta}_k - \tilde{\boldsymbol{\theta}}_k, -\nabla V(\boldsymbol{\theta}_k) + \nabla V(\tilde{\boldsymbol{\theta}}_k) \right\rangle$$

$$+ 2\eta\alpha \mathbb{E} \left\langle \boldsymbol{\theta}_k - \tilde{\boldsymbol{\theta}}_k, \phi[\frac{1}{M} \sum_{j=1}^{M} \delta_{\boldsymbol{\theta}_{k-jc/\eta}}](\boldsymbol{\theta}_k) - \phi[\pi_{M,c/\eta} * \tilde{\rho}_k](\tilde{\boldsymbol{\theta}}_k) \right\rangle$$

$$+ \eta^2 \mathbb{E} \left\| -\nabla V(\boldsymbol{\theta}_k) + \alpha\phi[\frac{1}{M} \sum_{j=1}^{M} \delta_{\boldsymbol{\theta}_{k-jc/\eta}}](\boldsymbol{\theta}_k) + \nabla V(\tilde{\boldsymbol{\theta}}_k) - \alpha\phi[\pi_{M,c/\eta} * \tilde{\rho}_k](\tilde{\boldsymbol{\theta}}_k) \right\|^2.$$

By the log-concavity, we have

$$\mathbb{E} \left\langle \boldsymbol{\theta}_k - \tilde{\boldsymbol{\theta}}_k, -\nabla V(\boldsymbol{\theta}_k) + \nabla V(\tilde{\boldsymbol{\theta}}_k) \right\rangle$$

$$\leq -L\mathbb{E} \left\| \boldsymbol{\theta}_k - \tilde{\boldsymbol{\theta}}_k \right\|^2,$$

for some positive constant $L$. And also, as $\eta$ is small, the last term on the right side of the equation is small term. Thus our main target is to bound the second term. We decompose the second term on the left side of the equation by

$$\mathbb{E} \left\langle \boldsymbol{\theta}_k - \tilde{\boldsymbol{\theta}}_k, \phi[\frac{1}{M} \sum_{j=1}^{M} \delta_{\boldsymbol{\theta}_{k-jc}}](\boldsymbol{\theta}_k) - \phi[\pi_{M,c/\eta} * \tilde{\rho}_k](\tilde{\boldsymbol{\theta}}_k) \right\rangle$$

$$= \mathbb{E} \left\langle \boldsymbol{\theta}_k - \tilde{\boldsymbol{\theta}}_k, \phi[\frac{1}{M} \sum_{j=1}^{M} \delta_{\boldsymbol{\theta}_{k-jc/\eta}}](\boldsymbol{\theta}_k) - \phi[\pi_{M,c/\eta} * \rho_k](\boldsymbol{\theta}_k) \right\rangle$$

$$+ \mathbb{E} \left\langle \boldsymbol{\theta}_k - \tilde{\boldsymbol{\theta}}_k, \phi[\pi_{M,c/\eta} * \rho_k](\boldsymbol{\theta}_k) - \phi[\pi_{M,c/\eta} * \tilde{\rho}_k](\boldsymbol{\theta}_k) \right\rangle$$

$$+ \mathbb{E} \left\langle \boldsymbol{\theta}_k - \tilde{\boldsymbol{\theta}}_k, \phi[\pi_{M,c/\eta} * \tilde{\rho}_k](\boldsymbol{\theta}_k) - \phi[\pi_{M,c/\eta} * \tilde{\rho}_k](\tilde{\boldsymbol{\theta}}_k) \right\rangle$$

$$= I_1 + I_2 + I_3.$$

We bound $I_1$, $I_2$ and $I_3$ independently.

**Bounding $I_1$** By Holder's inequality,

$$I_1 \leq \mathbb{E}\left[\left\|\boldsymbol{\theta}_k - \tilde{\boldsymbol{\theta}}_k\right\|\left\|\phi[\frac{1}{M}\sum_{j=1}^{M}\delta_{\boldsymbol{\theta}_{k-jc/\eta}}](\boldsymbol{\theta}_k) - \phi[\pi_{M,c/\eta} * \rho_k](\boldsymbol{\theta}_k)\right\|\right]$$

$$\leq \sqrt{\mathbb{E}\left\|\boldsymbol{\theta}_k - \tilde{\boldsymbol{\theta}}_k\right\|^2}\sqrt{\mathbb{E}\left\|\phi[\frac{1}{M}\sum_{j=1}^{M}\delta_{\boldsymbol{\theta}_{k-jc/\eta}}](\boldsymbol{\theta}_k) - \phi[\pi_{M,c/\eta} * \rho_k](\boldsymbol{\theta}_k)\right\|^2}.$$

We bound the second term on the right side of the inequality. Define

$$a_2 = \sup_k \frac{\mathbb{E}\left\|\phi[\frac{1}{M}\sum_{j=1}^{M}\delta_{\boldsymbol{\theta}_{k-jc/\eta}}](\boldsymbol{\theta}_k) - \phi[\pi_{M,c/\eta} * \rho_k](\boldsymbol{\theta}_k)\right\|^2}{\sup_{\|\boldsymbol{\theta}\|\leq B}\mathbb{E}\left\|\phi[\frac{1}{M}\sum_{j=1}^{M}\delta_{\boldsymbol{\theta}_{k-jc/\eta}}](\boldsymbol{\theta}) - \phi[\pi_{M,c/\eta} * \rho_k](\boldsymbol{\theta})\right\|^2}$$

and by the regularity assumption we know that

$$a_2 \leq \frac{\sup_k \mathbb{E}\left\|\phi[\frac{1}{M}\sum_{j=1}^{M}\delta_{\boldsymbol{\theta}_{k-jc}}](\boldsymbol{\theta}_k) - \phi[\pi_{M,c/\eta} * \rho_k](\boldsymbol{\theta}_k)\right\|^2}{\inf_k \sup_{\|\boldsymbol{\theta}\|\leq B}\mathbb{E}\left\|\phi[\frac{1}{M}\sum_{j=1}^{M}\delta_{\boldsymbol{\theta}_{k-jc/\eta}}](\boldsymbol{\theta}) - \phi[\pi_{M,c/\eta} * \rho_k](\boldsymbol{\theta})\right\|^2} < \infty.$$

Define $\phi[\frac{1}{M}\sum_{j=1}^{M}\delta_{\boldsymbol{\theta}_{k-jc/\eta}}](\boldsymbol{\theta}) - \phi[\pi_{M,c/\eta} * \rho_k](\boldsymbol{\theta}) = \phi^*[\frac{1}{M}\sum_{j=1}^{M}\delta_{\boldsymbol{\theta}_{k-jc/\eta}}]$ and since the stein operator is linear functional of the distribution, we have

$$\mathbb{E}\phi^*[\frac{1}{M}\sum_{j=1}^{M}\delta_{\boldsymbol{\theta}_{k-jc/\eta}}](\boldsymbol{\theta}) = 0,$$

given any $\boldsymbol{\theta}$. By Theorem B.1 that $\Theta_k$ is geometric ergodicity and thus is $\beta$-mixing with exponentially fast decay rate by Proposition B.2. And by Proposition B.1, we know that $\Theta_k$ is also $\alpha$-mixing with exponentially fast decay rate. We have the following estimation

$$\mathbb{E}\left\|\phi[\frac{1}{M}\sum_{j=1}^{M}\delta_{\boldsymbol{\theta}_{k-jc/\eta}}](\boldsymbol{\theta}_k) - \phi[\pi_{M,c/\eta} * \rho_k](\boldsymbol{\theta}_k)\right\|^2$$

$$\leq a_2 \sup_{\|\boldsymbol{\theta}\|\leq B}\mathbb{E}\left\|\phi^*[\frac{1}{M}\sum_{j=1}^{M}\delta_{\boldsymbol{\theta}_{k-jc/\eta}}](\boldsymbol{\theta})\right\|^2$$

$$\leq \frac{a_2}{M^2}\sup_{\|\boldsymbol{\theta}\|\leq B}\mathbb{E}\sum_{k=1}^{M}\left\|\phi^*[\delta_{\boldsymbol{\theta}_{t-kc/\eta}}](\boldsymbol{\theta})\right\|^2$$

$$+ \frac{a_2}{M^2}\sup_{\|\boldsymbol{\theta}\|\leq B}\mathbb{E}\sum_{k\neq j}\left\langle\phi^*[\delta_{\boldsymbol{\theta}_{t-kc/\eta}}](\boldsymbol{\theta}), \phi^*[\delta_{\boldsymbol{\theta}_{t-jc/\eta}}](\boldsymbol{\theta})\right\rangle$$

$$\leq \frac{Ca_2}{M}\left[\frac{e^{-rc}\left(1 - e^{-rMc}\right)}{1 - e^{rc}} + 1\right],$$

for some positive constant $r$ that characterize the decay rate of $\alpha$ mixing. Combine this two estimations, we have

$$I_1 \leq \sqrt{\mathbb{E}\left\|\boldsymbol{\theta}_k - \tilde{\boldsymbol{\theta}}_k\right\|^2}\sqrt{\frac{a_2 C}{M}\left[\frac{e^{-rc}\left(1 - e^{-rMc}\right)}{1 - e^{rc}} + 1\right]}.$$

**Bounding $I_2$** By Holder's inequality, we have

$$I_2 \leq \sqrt{\mathbb{E}\left\|\boldsymbol{\theta}_k - \tilde{\boldsymbol{\theta}}_k\right\|^2}\sqrt{\mathbb{E}\left\|\phi[\pi_{M,c/\eta} * \rho_k](\boldsymbol{\theta}_k) - \phi[\pi_{M,c/\eta} * \tilde{\rho}_k](\boldsymbol{\theta}_k)\right\|^2}.$$

We bound the second term in the right side of the inequality.

$$\mathbb{E}\left\|\phi[\pi_{M,c/\eta}*\rho_k](\boldsymbol{\theta}_k)-\phi[\pi_{M,c/\eta}*\tilde{\rho}_k](\boldsymbol{\theta}_k)\right\|^2$$

$$=\mathbb{E}\left\|\frac{1}{M}\sum_{j=1}^{M}\left[\phi[\rho_{k-jc/\eta}](\boldsymbol{\theta}_k)-\phi[\tilde{\rho}_{k-jc/\eta}](\boldsymbol{\theta}_k)\right]\right\|^2$$

$$\leq\frac{1}{M}\sum_{j=1}^{M}\mathbb{E}\left\|\phi[\rho_{k-jc/\eta}](\boldsymbol{\theta}_k)-\phi[\tilde{\rho}_{k-jc/\eta}](\boldsymbol{\theta}_k)\right\|^2$$

$$=\frac{1}{M}\sum_{j=1}^{M}\mathbb{E}_{\boldsymbol{\theta}_k}\left\|\mathbb{E}_{\boldsymbol{\theta}\sim\rho_{k-jc/\eta}}\bar{\phi}_{\boldsymbol{\theta}_k}(\boldsymbol{\theta})-\mathbb{E}_{\boldsymbol{\theta}\sim\tilde{\rho}_{k-jc/\eta}}\bar{\phi}_{\boldsymbol{\theta}_k}(\boldsymbol{\theta})\right\|^2$$

$$=\frac{1}{M}\sum_{j=1}^{M}\mathbb{E}_{\boldsymbol{\theta}_k}\sum_{i=1}^{d}\left|\mathbb{E}_{\boldsymbol{\theta}\sim\rho_{k-jc/\eta}}\bar{\phi}_{\boldsymbol{\theta}_k,i}(\boldsymbol{\theta})-\mathbb{E}_{\boldsymbol{\theta}\sim\tilde{\rho}_{k-jc/\eta}}\bar{\phi}_{\boldsymbol{\theta}_k,i}(\boldsymbol{\theta})\right|^2$$

$$\leq\frac{1}{M}\sum_{j=1}^{M}\mathbb{E}_{\boldsymbol{\theta}_k}\sum_{i=1}^{d}\left(\left\|\bar{\phi}_{\boldsymbol{\theta}_k,i}(\cdot)\right\|_{\mathcal{L}_\infty}\vee\left\|\bar{\phi}_{\boldsymbol{\theta}_k,i}(\cdot)\right\|_{\text{Lip}}\right)^2\mathbb{D}_{\text{BL}}^2\left[\rho_{k-jc/\eta},\tilde{\rho}_{k-jc/\eta}\right].$$

By Lemma B.4, we have

$$\sum_{i=1}^{d}\left(\left\|\bar{\phi}_{\boldsymbol{\theta}_k,i}(\cdot)\right\|_{\mathcal{L}_\infty}\vee\left\|\bar{\phi}_{\boldsymbol{\theta}_k,i}(\cdot)\right\|_{\text{Lip}}\right)^2$$

$$=\sum_{i=1}^{d}\left(\left\|\bar{\phi}_{\boldsymbol{\theta}_k,i}(\cdot)\right\|_{\mathcal{L}_\infty}^2\vee\left\|\bar{\phi}_{\boldsymbol{\theta}_k,i}(\cdot)\right\|_{\text{Lip}}^2\right)$$

$$\leq\left[4d\left\|\frac{2}{\sigma}e^{-\|\boldsymbol{\theta}\|^2/\sigma}\theta_1\right\|_{\text{BL}}^2+4\left\|e^{-\|\cdot\|^2/\sigma}\right\|_{\text{BL}}^2\|\nabla V(\boldsymbol{\theta}_k)\|^2\right].$$

Plug in the above estimation and by the relation that $\mathbb{D}_{\text{BL}}\leq\mathbb{W}_1\leq\mathbb{W}_2$, we have

$$\mathbb{E}\left\|\phi[\pi_{M,c}*\rho_k](\boldsymbol{\theta}_k)-\phi[\pi_{M,c}*\tilde{\rho}_k](\boldsymbol{\theta}_k)\right\|^2$$

$$\leq\left[4d\left\|\frac{2}{\sigma}e^{-\|\boldsymbol{\theta}\|^2/\sigma}\theta_1\right\|_{\text{BL}}^2+4\left\|e^{-\|\cdot\|^2/\sigma}\right\|_{\text{BL}}^2\mathbb{E}_{\boldsymbol{\theta}_k}\|\nabla V(\boldsymbol{\theta}_k)\|^2\right]\frac{1}{M}\sum_{j=1}^{M}\mathbb{D}_{\text{BL}}^2\left[\rho_{k-cj},\tilde{\rho_{k}-cj}\right]$$

$$\leq\left[4d\left\|\frac{2}{\sigma}e^{-\|\boldsymbol{\theta}\|^2/\sigma}\theta_1\right\|_{\text{BL}}^2+4\left\|e^{-\|\cdot\|^2/\sigma}\right\|_{\text{BL}}^2\mathbb{E}_{\boldsymbol{\theta}_k}\|\nabla V(\boldsymbol{\theta}_k)\|^2\right]\frac{1}{M}\sum_{j=1}^{M}\mathbb{W}_2^2\left[\rho_{k-cj},\tilde{\rho_{k}-cj}\right].$$

And combined all the estimation and by the definition of Wasserstein-distance, we conclude that

$$I_2\leq\sqrt{4d\left\|\frac{2}{\sigma}e^{-\|\boldsymbol{\theta}\|^2/\sigma}\theta_1\right\|_{\text{BL}}^2+4\left\|e^{-\|\cdot\|^2/\sigma}\right\|_{\text{BL}}^2\mathbb{E}_{\boldsymbol{\theta}_k}\|\nabla V(\boldsymbol{\theta}_k)\|^2}\sqrt{\frac{1}{M}\sum_{j=1}^{M}\mathbb{W}_2^2\left[\rho_{k-cj},\tilde{\rho}_{k-cj}\right]}$$

$$\leq\sqrt{4d\left\|\frac{2}{\sigma}e^{-\|\boldsymbol{\theta}\|^2/\sigma}\theta_1\right\|_{\text{BL}}^2+4\left\|e^{-\|\cdot\|^2/\sigma}\right\|_{\text{BL}}^2\mathbb{E}_{\boldsymbol{\theta}_k}\|\nabla V(\boldsymbol{\theta}_k)\|^2}\sqrt{\frac{1}{M}\sum_{j=1}^{M}\mathbb{E}\left\|\boldsymbol{\theta}_{k-cj}-\tilde{\boldsymbol{\theta}}_{k-cj}\right\|^2}.$$

**Bounding $I_3$** By Holder's inequality,

$$I_3\leq\sqrt{\mathbb{E}\left\|\boldsymbol{\theta}_k-\tilde{\boldsymbol{\theta}}_k\right\|^2}\sqrt{\mathbb{E}\left\|\phi[\pi_{M,c/\eta}*\tilde{\rho}_k](\boldsymbol{\theta}_k)-\phi[\pi_{M,c/\eta}*\tilde{\rho}_k](\tilde{\boldsymbol{\theta}}_k)\right\|^2}.$$

We bound the last term on the right side of the inequality. By assumption and Lemma B.3, we have

$$\mathbb{E}\left\|\phi[\pi_{M,c/\eta}*\tilde{\rho}_k](\boldsymbol{\theta}_k)-\phi[\pi_{M,c/\eta}*\tilde{\rho}_k](\tilde{\boldsymbol{\theta}}_k)\right\|^2$$

$$\leq\left[\left\|e^{-(\cdot)^2/\sigma}\right\|_{\text{Lip}}\mathbb{E}_{\boldsymbol{\theta}\sim\tilde{\rho}_k}\|\nabla V(\boldsymbol{\theta})\|+\left\|\frac{2}{\sigma}e^{-(\cdot)^2/\sigma}(\cdot)\right\|_{\text{Lip}}\right]^2\mathbb{E}\left\|\boldsymbol{\theta}_k-\tilde{\boldsymbol{\theta}}_k\right\|^2.$$

And combine the estimation, we have

$$I_3 \leq \left[ \left\| e^{-(\cdot)^2/\sigma} \right\|_{\text{Lip}} \mathbb{E}_{\boldsymbol{\theta} \sim \tilde{\rho}_k} \| \nabla V(\boldsymbol{\theta}) \| + \left\| \frac{2}{\sigma} e^{-(\cdot)^2/\sigma}(\cdot) \right\|_{\text{Lip}} \right] \mathbb{E} \left\| \boldsymbol{\theta}_k - \tilde{\boldsymbol{\theta}}_k \right\|^2 .$$

**Overall Bound**   Combine all the results, we have the following bound: for $k \geq Mc$,

$$\mathbb{E} \left\| \boldsymbol{\theta}_{k+1} - \tilde{\boldsymbol{\theta}}_{k+1} \right\|^2 - \mathbb{E} \left\| \boldsymbol{\theta}_k - \tilde{\boldsymbol{\theta}}_k \right\|^2$$

$$\leq - 2\eta L \mathbb{E} \left\| \boldsymbol{\theta}_k - \tilde{\boldsymbol{\theta}}_k \right\|^2$$

$$+ 2\eta\alpha \sqrt{\mathbb{E} \left\| \boldsymbol{\theta}_k - \tilde{\boldsymbol{\theta}}_k \right\|^2} \frac{c_1}{\sqrt{M}}$$

$$+ 2\eta\alpha c_2 \sqrt{\frac{1}{M} \sum_{j=1}^{M} \mathbb{E} \left\| \boldsymbol{\theta}_{k-jc/\eta} - \tilde{\boldsymbol{\theta}}_{k-jc/\eta} \right\|^2 \mathbb{E} \left\| \boldsymbol{\theta}_k - \tilde{\boldsymbol{\theta}}_k \right\|^2}$$

$$+ 2\eta\alpha c_3 \mathbb{E} \left\| \boldsymbol{\theta}_k - \tilde{\boldsymbol{\theta}}_k \right\|^2$$

$$+ \eta^2 c_4,$$

where

$$c_1 = \sqrt{a_2 C \left[ \frac{e^{-rc} \left( 1 - e^{-rMc} \right)}{1 - e^{rc}} + 1 \right]},$$

$$c_2 = \sqrt{4d \left\| \frac{2}{\sigma} e^{-\|\boldsymbol{\theta}\|^2/\sigma} \boldsymbol{\theta}_1 \right\|_{\text{BL}}^2 + 4 \left\| e^{-\|\cdot\|^2/\sigma} \right\|_{\text{BL}}^2 \sup_k \mathbb{E}_{\boldsymbol{\theta}_k} \| \nabla V(\boldsymbol{\theta}_k) \|^2},$$

$$c_3 = \left[ \left\| e^{-(\cdot)^2/\sigma} \right\|_{\text{Lip}} \sup_k \mathbb{E}_{\boldsymbol{\theta} \sim \tilde{\rho}_k} \| \nabla V(\boldsymbol{\theta}) \| + \left\| \frac{2}{\sigma} e^{-(\cdot)^2/\sigma}(\cdot) \right\|_{\text{Lip}} \right],$$

and

$$c_4 = \sup_{k \geq Mc/\eta} \mathbb{E} \left\| \nabla V(\boldsymbol{\theta}_k) + \alpha\phi[\frac{1}{M} \sum_{j=1}^{M} \delta_{\boldsymbol{\theta}_{k-jc/\eta}}](\boldsymbol{\theta}_k) - \nabla V(\tilde{\boldsymbol{\theta}}_k) - \alpha\phi[\pi_{M,c/\eta} * \tilde{\rho}_k](\tilde{\boldsymbol{\theta}}_k) \right\|^2 .$$

Define $u_k = \sqrt{\mathbb{E} \left\| \boldsymbol{\theta}_k - \tilde{\boldsymbol{\theta}}_k \right\|^2}$ and $U_k = \sup_{l \in [k]} u_l$, we have

$$U_{k+1}^2 \leq q U_k^2 + \frac{2\eta\alpha c_1}{\sqrt{M}} U_k + \eta^2 c_4,$$

where $q = (1 - 2\eta(L - \alpha c_2 - \alpha c_3))$. By the assumption that $\alpha \leq L/(c_2 + c_3)$, $q < 1$. Now we prove the bound of $U_k$ by induction. We take the hypothesis that $U_k^2 \leq \frac{\left( \frac{2\eta\alpha c_1}{\sqrt{M}} + (1-q)\eta(c_4 + \frac{1}{1-q}) \right)^2}{(1-q)^2}$

and notice that the hypothesis holds for $U_0 = 0$. By the hypothesis, we have

$$U_{k+1}^2 \leq q \frac{\left(\frac{2\eta\alpha c_1}{\sqrt{M}} + (1-q)\eta\left(c_4 + \frac{1}{1-q}\right)\right)^2}{(1-q)^2} + \frac{2\eta\alpha c_1}{\sqrt{M}} \frac{\left(\frac{2\eta\alpha c_1}{\sqrt{M}} + (1-q)\eta\left(c_4 + \frac{1}{1-q}\right)\right)}{(1-q)} + \eta^2\left(c_4 + \frac{1}{1-q}\right)$$

$$= q \frac{\left(\frac{2\eta\alpha c_1}{\sqrt{M}} + (1-q)\eta\left(c_4 + \frac{1}{1-q}\right)\right)^2}{(1-q)^2} + \frac{1}{1-q}\left(\frac{2\eta\alpha c_1}{\sqrt{M}}\right)^2 + \frac{2\eta\alpha c_1}{\sqrt{M}}\eta\left(c_4 + \frac{1}{1-q}\right) + \eta^2\left(c_4 + \frac{1}{1-q}\right)$$

$$= q \frac{\left(\frac{2\eta\alpha c_1}{\sqrt{M}} + (1-q)\eta\left(c_4 + \frac{1}{1-q}\right)\right)^2}{(1-q)^2}$$
$$+ \frac{1-q}{(1-q)^2}\left[\left(\frac{2\eta\alpha c_1}{\sqrt{M}}\right)^2 + (1-q)\frac{2\eta\alpha c_1}{\sqrt{M}}\eta\left(c_4 + \frac{1}{1-q}\right) + (1-q)\eta^2\left(c_4 + \frac{1}{1-q}\right)\right]$$

$$\leq q \frac{\left(\frac{2\eta\alpha c_1}{\sqrt{M}} + (1-q)\eta\left(c_4 + \frac{1}{1-q}\right)\right)^2}{(1-q)^2}$$
$$+ \frac{1-q}{(1-q)^2}\left[\left(\frac{2\eta\alpha c_1}{\sqrt{M}}\right)^2 + (1-q)\frac{2\eta\alpha c_1}{\sqrt{M}}\eta\left(c_4 + \frac{1}{1-q}\right) + (1-q)^2\eta^2\left(c_4 + \frac{1}{1-q}\right)^2\right]$$

$$= q \frac{\left(\frac{2\eta\alpha c_1}{\sqrt{M}} + (1-q)\eta\left(c_4 + \frac{1}{1-q}\right)\right)^2}{(1-q)^2} + (1-q)\frac{\left(\frac{2\eta\alpha c_1}{\sqrt{M}} + (1-q)\eta\left(c_4 + \frac{1}{1-q}\right)\right)^2}{(1-q)^2}$$

$$= \frac{\left(\frac{2\eta\alpha c_1}{\sqrt{M}} + (1-q)\eta\left(c_4 + \frac{1}{1-q}\right)\right)^2}{(1-q)^2},$$

where the last second inequality holds by $(1-q)\left(c_4 + \frac{1}{1-q}\right) \geq 1$. Thus we complete the argument of induction and we have, for any $k$,

$$U_k^2 \leq \frac{\left(\frac{2\eta\alpha c_1}{\sqrt{M}} + (1-q)\eta\left(c_4 + \frac{1}{1-q}\right)\right)^2}{(1-q)^2}$$

$$\leq 2\frac{\frac{4\eta^2\alpha^2 c_1^2}{M} + 2(1-q)^2\eta^2\left(c_4 + \frac{1}{1-q}\right)^2}{(1-q)^2}$$

$$= \frac{2\alpha^2 c_1^2}{(L - \alpha c_2 - \alpha c_3)^2}\frac{1}{M} + 4\eta^2\left(c_4 + 2\eta(L - \alpha c_2 - \alpha c_3)\right)^2.$$

And it implies that $\mathbb{W}_2^2[\rho_k, \tilde{\rho}_k] \leq u_k \leq U_k \leq \frac{2\alpha^2 c_1^2}{(L-\alpha c_2-\alpha c_3)^2}\frac{1}{M} + 4\eta^2\left(c_4 + 2\eta(L - \alpha c_2 - \alpha c_3)\right)^2$.

### B.6 PROOF OF TECHNICAL LEMMAS

#### B.6.1 PROOF OF LEMMA B.1

For the first part:

$$\|\nabla V(\boldsymbol{\theta})\|$$
$$\leq \|\nabla V(\boldsymbol{\theta}) - \nabla V(\mathbf{0})\| + \|\nabla V(\mathbf{0})\|$$
$$\leq b_1\left(\|\boldsymbol{\theta}_1\| + 1\right).$$

For the second part:

$$
\begin{aligned}
&\|\boldsymbol{\theta} - \eta\nabla V(\boldsymbol{\theta})\| \\
&= \langle \boldsymbol{\theta} - \eta\nabla V(\boldsymbol{\theta}), \boldsymbol{\theta} - \eta\nabla V(\boldsymbol{\theta}) \rangle \\
&= \|\boldsymbol{\theta}\|^2 + 2\eta\langle \boldsymbol{\theta}, -\nabla V(\boldsymbol{\theta}) \rangle + \eta^2\|\nabla V(\boldsymbol{\theta})\|^2 \\
&\leq \|\boldsymbol{\theta}\|^2 + 2\eta\left(-a_1\|\boldsymbol{\theta}\|^2 + b_1\right) + \eta^2 b_1(1 + \|\boldsymbol{\theta}\|^2) \\
&= \left(1 - 2\eta a_1 + \eta^2 b_1\right)\|\boldsymbol{\theta}\|^2 + \eta^2 b_1 + 2\eta b_1.
\end{aligned}
$$

### B.6.2 PROOF OF LEMMA B.2

It is obvious that $\|K\|_{\infty,\infty} \leq 1$.

$$
\begin{aligned}
&\|K(\boldsymbol{\theta}', \boldsymbol{\theta}_1) - K(\boldsymbol{\theta}', \boldsymbol{\theta}_2)\| \\
&\left\| e^{-\|\boldsymbol{\theta}'-\boldsymbol{\theta}_1\|^2/\sigma} - e^{-\|\boldsymbol{\theta}'-\boldsymbol{\theta}_2\|^2/\sigma} \right\| \\
&\leq \left\| e^{-(\cdot)^2/\sigma} \right\|_{\mathrm{Lip}} \|\boldsymbol{\theta}_1 - \boldsymbol{\theta}_2\|_2.
\end{aligned}
$$

And

$$
\begin{aligned}
&\|\nabla_{\boldsymbol{\theta}'} K(\boldsymbol{\theta}', \boldsymbol{\theta}_1) - \nabla_{\boldsymbol{\theta}'} K(\boldsymbol{\theta}', \boldsymbol{\theta}_2)\| \\
&= \left\| \frac{2}{\sigma} e^{-\|\boldsymbol{\theta}'-\boldsymbol{\theta}_1\|^2/\sigma}(\boldsymbol{\theta}' - \boldsymbol{\theta}_1) - \frac{2}{\sigma} e^{-\|\boldsymbol{\theta}'-\boldsymbol{\theta}_2\|^2/\sigma}(\boldsymbol{\theta}' - \boldsymbol{\theta}_2) \right\| \\
&\leq \left\| \frac{2}{\sigma} e^{-(\cdot)^2/\sigma}(\cdot) \right\|_{\mathrm{Lip}} \|\boldsymbol{\theta}_1 - \boldsymbol{\theta}_2\|_2.
\end{aligned}
$$

### B.6.3 PROOF OF LEMMA B.3

For any distribution $\rho$ such that $\int \|\nabla_{\boldsymbol{\theta}} V(\boldsymbol{\theta})\| \rho(\boldsymbol{\theta}) d\boldsymbol{\theta} < \infty$,

$$
\begin{aligned}
&\|\phi[\rho](\boldsymbol{\theta}_1) - \phi[\rho](\boldsymbol{\theta}_2)\| \\
&= \|\mathbb{E}_{\boldsymbol{\theta}\sim\rho}\left\{-\left[K(\boldsymbol{\theta}, \boldsymbol{\theta}_1) - K(\boldsymbol{\theta}, \boldsymbol{\theta}_2)\right]\nabla V(\boldsymbol{\theta}) + \nabla_1 K(\boldsymbol{\theta}, \boldsymbol{\theta}_1) - \nabla_1 K(\boldsymbol{\theta}, \boldsymbol{\theta}_2)\right\}\| \\
&\leq \left\| e^{-(\cdot)^2/\sigma} \right\|_{\mathrm{Lip}} \mathbb{E}_{\boldsymbol{\theta}\sim\rho}\|\nabla V(\boldsymbol{\theta})\| \|\boldsymbol{\theta}_1 - \boldsymbol{\theta}_2\|_2 \\
&+ \left\| \frac{2}{\sigma} e^{-(\cdot)^2/\sigma}(\cdot) \right\|_{\mathrm{Lip}} \|\boldsymbol{\theta}_1 - \boldsymbol{\theta}_2\|_2.
\end{aligned}
$$

For proving the second result, we notice that

$$
\begin{aligned}
\|\phi[\rho](\boldsymbol{\theta})\| &= \mathbb{E}_{\boldsymbol{\theta}'\sim\rho}\left[K(\boldsymbol{\theta}', \boldsymbol{\theta})\nabla V(\boldsymbol{\theta}') + \nabla_1 K(\boldsymbol{\theta}', \boldsymbol{\theta})\right] \\
&\leq \|K\|_\infty \mathbb{E}_{\boldsymbol{\theta}'\sim\rho}\left[\|\nabla V(\boldsymbol{\theta}')\| + \frac{2}{\sigma}\left(\|\boldsymbol{\theta}'\| + \|\boldsymbol{\theta}\|\right)\right] \\
&\leq \|K\|_\infty b_1 + \mathbb{E}_{\boldsymbol{\theta}'\sim\rho}\left[\left(\frac{2}{\sigma} + b_1\right)\|\boldsymbol{\theta}'\| + \|\boldsymbol{\theta}\|\right].
\end{aligned}
$$

### B.6.4 PROOF OF LEMMA B.4

Given any $\boldsymbol{\theta}'$,

$$\sum_{i=1}^{d}\left\|\bar{\phi}_{\boldsymbol{\theta}',i}(\boldsymbol{\theta})\right\|_{\mathrm{Lip}}^{2}$$

$$=\sum_{i=1}^{d}\left[\sup_{\boldsymbol{\theta}_1\neq\boldsymbol{\theta}_2}\frac{\left|\bar{\phi}_{\boldsymbol{\theta}',i}(\boldsymbol{\theta}_1)-\bar{\phi}_{\boldsymbol{\theta}',i}(\boldsymbol{\theta}_2)\right|}{\left\|\boldsymbol{\theta}_1-\boldsymbol{\theta}_2\right\|_{2}}\right]^{2}$$

$$=\sum_{i=1}^{d}\sup_{\boldsymbol{\theta}_1\neq\boldsymbol{\theta}_2}\frac{\left|\bar{\phi}_{\boldsymbol{\theta}',i}(\boldsymbol{\theta}_1)-\bar{\phi}_{\boldsymbol{\theta}',i}(\boldsymbol{\theta}_2)\right|^{2}}{\left\|\boldsymbol{\theta}_1-\boldsymbol{\theta}_2\right\|_{2}^{2}}$$

$$\leq 2\sum_{i=1}^{d}\sup_{\boldsymbol{\theta}_1\neq\boldsymbol{\theta}_2}\frac{\left|\left(e^{-\left\|\boldsymbol{\theta}'-\boldsymbol{\theta}_1\right\|^{2}/\sigma}-e^{-\left\|\boldsymbol{\theta}'-\boldsymbol{\theta}_2\right\|^{2}/\sigma}\right)\frac{\partial}{\partial\theta'_i}V(\boldsymbol{\theta}')\right|^{2}}{\left\|\boldsymbol{\theta}_1-\boldsymbol{\theta}_2\right\|_{2}^{2}}$$

$$+2\sum_{i=1}^{d}\sup_{\boldsymbol{\theta}_1\neq\boldsymbol{\theta}_2}\frac{\left|\frac{2}{\sigma}e^{-\left\|\boldsymbol{\theta}'-\boldsymbol{\theta}_1\right\|^{2}/\sigma}(\boldsymbol{\theta}_{1,i}-\boldsymbol{\theta}'_i)-\frac{2}{\sigma}e^{-\left\|\boldsymbol{\theta}'-\boldsymbol{\theta}_2\right\|^{2}/\sigma}(\boldsymbol{\theta}_{2,i}-\boldsymbol{\theta}'_i)\right|^{2}}{\left\|\boldsymbol{\theta}_1-\boldsymbol{\theta}_2\right\|_{2}^{2}}.$$

For the first term on the right side of the inequality,

$$\sum_{i=1}^{d}\sup_{\boldsymbol{\theta}_1\neq\boldsymbol{\theta}_2}\frac{\left|\left(e^{-\left\|\boldsymbol{\theta}'-\boldsymbol{\theta}_1\right\|^{2}/\sigma}-e^{-\left\|\boldsymbol{\theta}'-\boldsymbol{\theta}_2\right\|^{2}/\sigma}\right)\frac{\partial}{\partial\theta'_i}V(\boldsymbol{\theta}')\right|^{2}}{\left\|\boldsymbol{\theta}_1-\boldsymbol{\theta}_2\right\|_{2}^{2}}$$

$$=\sum_{i=1}^{d}\left|\frac{\partial}{\partial\theta_i}V(\boldsymbol{\theta}')\right|^{2}\sup_{\boldsymbol{\theta}_1\neq\boldsymbol{\theta}_2}\frac{\left|\left(e^{-\left\|\boldsymbol{\theta}'-\boldsymbol{\theta}_1\right\|^{2}/\sigma}-e^{-\left\|\boldsymbol{\theta}'-\boldsymbol{\theta}_2\right\|^{2}/\sigma}\right)\right|^{2}}{\left\|\boldsymbol{\theta}_1-\boldsymbol{\theta}_2\right\|_{2}^{2}}$$

$$=\left\|\nabla V(\boldsymbol{\theta}')\right\|^{2}\left\|e^{-\|\cdot\|^{2}/\sigma}\right\|_{\mathrm{Lip}}^{2}.$$

To bound the second term, by the symmetry of each coordinates, we have

$$\sum_{i=1}^{d}\sup_{\boldsymbol{\theta}_1\neq\boldsymbol{\theta}_2}\frac{\left|\frac{2}{\sigma}e^{-\left\|\boldsymbol{\theta}'-\boldsymbol{\theta}_1\right\|^{2}/\sigma}(\theta_{1,i}-\theta'_i)-\frac{2}{\sigma}e^{-\left\|\boldsymbol{\theta}'-\boldsymbol{\theta}_2\right\|^{2}/\sigma}(\theta_{1,i}-\theta'_i)\right|^{2}}{\left\|\boldsymbol{\theta}_1-\boldsymbol{\theta}_2\right\|_{2}^{2}}$$

$$=d\left\|\frac{2}{\sigma}e^{-\|\boldsymbol{\theta}\|^{2}/\sigma}\theta_1\right\|_{\mathrm{Lip}}^{2}.$$

This finishes the first part of the lemma.

$$\sum_{i=d}^{d}\left\|\bar{\phi}_{\boldsymbol{\theta}',i}(\boldsymbol{\theta})\right\|_{\mathcal{L}_\infty}^{2}$$

$$=\sum_{i=d}^{d}\left\|e^{-\left\|\boldsymbol{\theta}'-\boldsymbol{\theta}\right\|^{2}/\sigma}\left(\frac{2}{\sigma}\boldsymbol{\theta}_i-\frac{2}{\sigma}\boldsymbol{\theta}'_i-\frac{\partial}{\partial\theta'_i}V(\boldsymbol{\theta}')\right)\right\|_{\mathcal{L}_\infty}^{2}$$

$$\leq\sum_{i=d}^{d}2\left\|\frac{2}{\sigma}e^{-\left\|\boldsymbol{\theta}'-\boldsymbol{\theta}\right\|^{2}/\sigma}(\boldsymbol{\theta}_i-\boldsymbol{\theta}'_i)\right\|_{\mathcal{L}_\infty}^{2}+\sum_{i=d}^{d}2\left\|e^{-\left\|\boldsymbol{\theta}'-\boldsymbol{\theta}\right\|^{2}/\sigma}\frac{\partial}{\partial\theta'_i}V(\boldsymbol{\theta}')\right\|_{\mathcal{L}_\infty}^{2}$$

$$\leq\sum_{i=d}^{d}2\left\|\frac{2}{\sigma}e^{-\left\|\boldsymbol{\theta}'-\boldsymbol{\theta}\right\|^{2}/\sigma}(\boldsymbol{\theta}_i-\boldsymbol{\theta}'_i)\right\|_{\mathcal{L}_\infty}^{2}+2\left\|e^{-\|\cdot\|^{2}/\sigma}\right\|_{\mathcal{L}_\infty}^{2}\left\|\nabla V(\boldsymbol{\theta}')\right\|^{2}$$

$$\leq 2d\left\|\frac{2}{\sigma}e^{-\|\boldsymbol{\theta}\|^{2}/\sigma}\theta_1\right\|_{\mathcal{L}_\infty}^{2}+2\left\|e^{-\|\cdot\|^{2}/\sigma}\right\|_{\mathcal{L}_\infty}^{2}\left\|\nabla V(\boldsymbol{\theta}')\right\|^{2}.$$

