# OpenReview forum: "Stein Self-Repulsive Dynamics: Benefits from Past Samples"
_ICLR.cc/2020/Conference — Reject_

### Official Review · AnonReviewer1 · 2019-10-20
**Official Blind Review #1**

**Rating:** 3

**Review:**

The papers described how to use the repulsive term used with standard SVGD within MCMC/SGLD. Briefly, the paper proposes to use a (damped version of) the SVG repulsive term between the current position of a SGLD trajectory and the empirical distribution defined by the trajectory.

The approach is interesting and natural. Unfortunately, I do not think that the experiments are convincing.

(1) Since the authors are advertising the Bayesian framework, the choices of metrics such as test RMSE or test LL are not adapted
(2) in the UCI dataset examples, it is indeed extremely difficult to explain the enhanced performances? Is it because of multimodality? Better exploration of a mode? Comparison to a single NN? Comparison with an ensemble of NN?
(3) I would have been much more convinced by a set of well-chosen and controlled experiments. The 2-dimensional examples are far too low-dimensional to be convincing. Higher-dimensional Gaussian? Higher-dimensional mixtures? Non-linear tractable problems in higher dimensions? Influence of the tuning parameters (RBF parameter? alpha? step-size in SGLD)? Computational issues? Subsampling-effect? etc...

The proposed method is interesting and has a lot of potential. I would like to suggest the authors to spend more times on careful and controlled numerical experiments (Bayesian NN are not very good for this purpose) -- with convincing numerics (which would give more reasons to delve into the proofs) the method can be very promising.


**Experience Assessment:**

I have published in this field for several years.

**Review Assessment: Checking Correctness Of Derivations And Theory:**

I did not assess the derivations or theory.

**Review Assessment: Checking Correctness Of Experiments:**

I carefully checked the experiments.

**Review Assessment: Thoroughness In Paper Reading:**

I read the paper at least twice and used my best judgement in assessing the paper.

---

> ### Author Response · Authors · 2019-11-09
> **Thanks for your suggestion on experiments. Below are our responses. (1/n)**
>
> Thanks very much for your suggestion on the experiments design. We respectfully disagree with criticism on our experiments. As we elaborate below, we believe our experiments are conducted properly and convincingly, with the same standard as many published works in this field. Meanwhile, following your suggestion, we also conduct extra experiments to support the proposed method (See A2. and A3, Appendix).
>
> In addition, we also encourage the reviewer to focus more on our algorithmic and theoretical contribution, which is the main highlight of the paper.
>
> ----------------------------------------------------------------------------------------------------------------------------
>
> Q1: Since the authors are advertising the Bayesian framework, the choices of metrics such as test RMSE or test LL are not adapted.
>
> A1: We disagree with your statement. As far as we know,  it is a common practice to use test RMSE and test LL as evaluation metrics in papers on Bayesian methods in the machine learning communities. See, for example, SGHMC[1], SGNHT [2] in the sampling literature as well as SVGD [3], variational inference with normalizing flow [4], KIVI [5], in the variational inference literature. So we believe the metrics like test RMSE or test LL can be adapted in Bayesian setting.
>
> ----------------------------------------------------------------------------------------------------------------------------
>
> Q2: In the UCI dataset examples, it is indeed extremely difficult to explain the enhanced performances? Is it because of multimodality? Better exploration of a mode? Comparison to a single NN? Comparison with an ensemble of NN?
>
> A2:  It is hard to interpret the gain for sure, as the dimension is high. We conjecture the gains is from the better exploration of the posterior domain, which matches our intuition. The reasons are as follows:
>
>     1. Just as Figure 1, 4, 6 shows, our methods can better explore the under-explored region given the previous samples, so we expect similar behaviors in the higher dimensional setting.
>
>     2. In the contextual bandits experiment, our method is significantly better than Langevin. This is because in this problem, a better uncertainty estimation is the key for the performance as described in [8, 9, 10]. And we think the better performance in the contextual bandits experiment support our idea on better exploration of the posterior domain.
>
>     3. We also provide more detailed experiment on the Appendix (e.g. Figure 7, 8 and table 3, 4). With growing number of samples, the proposed SRLD decreases the test RMSE and increases the test LL faster than LD, which is due to the better exploration of posterior domain.
>
> Regarding the multimodality question, it’s known that neural networks suffer from multimodality, e.g. in [6] the authors discussed the multimodality of neural network, but this is beyond the scope of our paper.
>
> Regarding the comparison with a single NN or an ensemble of NN, we think it’s not very meaningful to compare the result with them. Because our purpose is to test the performance of different sampling methods but single or ensemble of neural networks are not designed for sampling the posterior of Bayesian Neural Networks.
>
> In the end, we want to highlight that UCI examples have become a standard benchmark in this area as we mentioned in A1, and we believe it gives good quantification on sampling quality.

---

> ### Author Response · Authors · 2019-11-09
> **Thanks for your suggestion on experiments. Below are our responses. (2/n)**
>
> Q3: I would have been much more convinced by a set of well-chosen and controlled experiments. The 2-dimensional examples are far too low-dimensional to be convincing. Higher-dimensional Gaussian? Higher-dimensional mixtures? Non-linear tractable problems in higher dimensions? Influence of the tuning parameters (RBF parameter? alpha? step-size in SGLD)? Computational issues? Subsampling-effect?
>
> A3: We would like to point out that the main aim of the low dimensional toy example is to transparently visualize how the proposed repulsive force helps sampling in a more intuitive way. We believe it is an necessary and informative experiment. We also found several other papers also use this way to build the intuition, e.g. [1, 3, 5].
>
> Following your suggestion, we conduct additional Higher dimensional Gaussian experiment with d = 100 and Gaussian mixture experiment with dimension = 20 with different values of alpha. The result and description of the experiment is at Appendix A.2 and A.3.
>
> We also believe that the experiment are well-controlled: In all of the 2D toy experiments, we use the update norm to rule out the influence of additional gradient. In all the experiment, we couple SRLD with LD by using the same initialization and same sequence of Gaussian noise in each repeat.
>
> Regarding the influence of tuning parameters, since it is not realistic to do ablation study on all the hyper-parameter (and many of them are not introduced by this paper), we simply fix many of them and tune the step size for each sampling method in the same way. We believe this is a standard experiment setting that is broadly used in the machine learning community, e.g. [1, 2, 3, 4, 5, 6, 7] etc. The setting of other hyper-parameters is as follows (as we mentioned in the paper):
>
>     1.RBF parameter: As described in the paper, we use the median heuristics introduced in [3] to choose the bandwidth of RBF kernel, which is a generally used heuristics (and also a well accepted setting in the community) for kernel-based variational inference methods, for example, papers [5, 6, 7] also use this trick without more ablation study.
>
>      2.Alpha: As mentioned in the paper, we simply fix this parameter without further tuning. We believe further tuning of this parameter can further improve the quality of the proposed method. We study the effect of alpha in the new additional experiment in A.2 and A.3 appendix.
>
>     3.Number of mini-batch samples: We simply use the same number of mini-batch samples for each method in each experiment as [1, 2, 3, 4, 5] etc. do.
>
> For the computational issue, we just need additional O(M) time to calculate the repulsive gradient, which is not expensive, given the fact that current libraries are well optimized to do matrix multiplication/addition.
>
> [1] Chen, Tianqi, Emily Fox, and Carlos Guestrin. “Stochastic gradient hamiltonian monte carlo." International conference on machine learning. 2014.
> [2] Ding, Nan, et al. "Bayesian sampling using stochastic gradient thermostats." Advances in neural information processing systems. 2014.
> [3] Liu, Qiang, and Dilin Wang. "Stein variational gradient descent: A general purpose bayesian inference algorithm." Advances in neural information processing systems. 2016.
> [4] Rezende, Danilo, and Shakir Mohamed. "Variational Inference with Normalizing Flows." International Conference on Machine Learning. 2015.
> [5] Shi, Jiaxin, Sun, Shengyang and Zhu, Jun. “Kernel implicit variational inference”, International Conference on Learning Representations. 2018.
> [6] Wang, Ziyu, et al. "Function Space Particle Optimization for Bayesian Neural Networks." International Conference on Learning Representations. 2019.
> [7] Chen, Changyou, et al. “A Unified Particle-Optimization Framework for Scalable Bayesian Sampling” Conference on Uncertainty in Artificial Intelligence. 2018
> [8] Bubeck, Sébastien, and Nicolo Cesa-Bianchi. "Regret analysis of stochastic and nonstochastic multi-armed bandit problems." Foundations and Trends® in Machine Learning 5.1 (2012): 1-122.
> [9] Riquelme, Carlos et al. “Deep Bayesian Bandits Showdown: An Empirical Comparison of Bayesian Deep Networks for Thompson Sampling.” International Conference on Learning
> Representations. 2018.
> [10] May, Benedict C., et al. "Optimistic Bayesian sampling in contextual-bandit problems." Journal of Machine Learning Research 13.Jun (2012): 2069-2106.

---

### Official Review · AnonReviewer3 · 2019-10-23
**Official Blind Review #3**

**Rating:** 6

**Review:**

This paper proposed another variant of Langevin dynamics, called “Stein self-repulsive dynamics,” which simultaneously decreases the auto-correlation of Langevin dynamics and eliminates the need for running parallel chains in SVGD. They combined Langevin dynamics with Stein variational gradient descent and theoretically justified that the proposed method successfully converges to the stationary distribution with only a single chain, unlike SVGD. The proposed method decreases the auto-correlation of Langevin dynamics, so the proposed method increases the sample efficiency.

The paper is well-written. The idea of the proposed method is natural, which is incorporating the functionality of SVGD to reduce the auto-correlation of Langevin dynamics. The idea is intuitive and justified by their theoretical analysis. The authors also well- placed their work in the literature, as described in Section 3. The intuitive explanation of the proposed method is given in Section 3.

I have one technical question as follows. If the authors reply appropriately, I will raise the score to accept.

In Theorem 4.3, the result holds for any k and M. The authors claim that if we take a limit of M -> ∞ with fixed k, the practical dynamics converges to the discrete-time mean-field limit, in Section 4. However, to state the result of Theorem 4.3, k should be bigger than M c_\eta from the dentition of \tilde{\rho}_k^M, as shown under the equation (4). How do we take a limit of M -> ∞ ? Does k also go ∞?

Minor comments:
- The definition of g should depend on only \theta_k^I and \hat{\delta}_k^M, not \theta_k^k.
- The equation (1) should hold for any \theta’, not \theta.
- The equation (1) should contain \rho, not p.

**Experience Assessment:**

I have published one or two papers in this area.

**Review Assessment: Checking Correctness Of Derivations And Theory:**

I assessed the sensibility of the derivations and theory.

**Review Assessment: Checking Correctness Of Experiments:**

I assessed the sensibility of the experiments.

**Review Assessment: Thoroughness In Paper Reading:**

I read the paper at least twice and used my best judgement in assessing the paper.

---

> ### Author Response · Authors · 2019-11-09
> **Thanks for your review! Below are our responses.**
>
> Q1: In Theorem 4.3, the result holds for any $k$ and $M$. The authors claim that if we take a limit of $M \to \infty$ with fixed $k$, the practical dynamics converges to the discrete-time mean-field limit, in Section 4. However, to state the result of Theorem 4.3, $k$ should be bigger than $M c_\eta$ from the dentition of $\tilde{\rho}_k^M$, as shown under the equation (4). How do we take a limit of $M \to\infty$ ? Does k also go $\infty$?
>
> A1: Thanks for pointing this out. The result of this theorem holds uniformly for any $k$ (not a fixed $k$). Besides, we do not require $k$ bigger than $M c_\eta$ in the definition of $\tilde{\rho}_k^M$. When $k$ is no more than $M c_\eta$, $\tilde{\rho}_k^M$ and $\rho_k^M$ are stochastic processes with same distribution and thus the Wasserstein distance between them is 0. And for any $k$ is greater than $M c_\eta$, we have the uniform bound (w.r.t. $k$) as stated in the theorem 4.3. We are sorry for not stating this clearly in the theorem and we have revisited the present of the theorem. We will fix this issue in the next revision.
>
> We also point out that, as our system is complicated, in taking the limit of $M\to\infty$, we need to ensure that the number of iteration we run is larger than $Mc_\eta$. To be specific, the asymptotic convergence would be
>
> $$\lim_{k,M \to\infty, \eta \to 0^+} \mathbb{D}_{\text{BL}} (\rho_k, \rho^*)=0$$
>
> where the joint limit of k and M requires that $k\eta\to\infty$; $\exp(C\alpha^{2}k\eta)\eta^{2}=o(1)$; $(k\eta)/(Mc)=q(1+o(1))$ with $q>1$. Here if $q \leq 1$, we degenerate to Langevin. But when $q>1$ (intuitively that means, when $M$ is large, the number of iterations we run is larger), our dynamics is different from Langevin, which is what we do in the practice.
>
> Also, we would like to remark that this seemingly strange things is in fact the ‘artifact’ caused by the using of Langevin dynamics at beginning to obtain the $M$ initial samples when we designed the practical implementation of the proposed methods. However, it is not really necessary to use Langevin dynamics to get $M$ initial samples, as we can simply using some other initialization distribution and get the $M$ initial samples from that distribution (and by this setting, our dynamics is simply the second phases in Eq (3)). All our theory can be easily generalized to this setting using almost identical argument, which can also address your concerns on this issue.
>
> Q2: Regarding other minor comments
> A2: Thanks for your notification! We will polish our paper and rewrite the corresponding part in the next revision.

---

### Official Review · AnonReviewer2 · 2019-10-26
**Official Blind Review #2**

**Rating:** 6

**Review:**

I have the rebuttal of the authors, the paper improved indeed and some point on role of M is better clarified now although it is still a bit convoluted. The paper would be stronger if the analysis shows any theoretical advantage to the presented method. I think the author put a good effort in addressing some of my concerns and I m raising my score to 6.


####
Summary of the paper:

The paper proposes stein self repulsive dynamics for sampling from an unnormalized distribution. The method starts by using Langevin dynamic for up to time $Mc$ and then uses those pasts samples to guide the trajectories of the langevin sampling to explore new areas of the densities  using the stein witness function between the current particles  and the past samples( similar to Stein Variational gradient descent).

The paper analyses the mixing properties under standard assumptions of the potential of the Boltzmann distribution and the kernel used in the Stein discrepancy, and shows convergence to the boltzman distribution as the number of particles goes to infinity and the step size goes to zero.

Authors validate their methods and show that it indeed explores on a synthetic example new areas wrt to pure langevin dynamics. Applications in sampling from the posterior of bayesian neural networks and contextual bandits compare the performance of the proposed method to langevin dynamic and pure stein descent favorably.

Clarity/presentation :

The paper is well written and the intuition are well presented .

The notation $\hat{delta}_{M} $ is not great in denoting direct measures, please using another symbol.

My main concerns with the paper are the following:

- the definition of $\bar{\delta}_{M}$ averages only $M$ particles choosen for $M$ time stamps. Something is off here for the continuous approximation to work at each time stamp you need $N$ particles and then you have a past horizon $M$. As $M\to \infty$ this does not matter, but I think in your implementation you are considering at each time step $N$ particles and you average on a horizon of size $M$. is this correct? Please clarify?

- The theorem show only asymptotic behavior and don't quantify the intuition behind the paper , that the "coverage of the samples" is higher.
 Can you for instance bound the wasserstein  distance between the pure langevin and Stein Repulsive dynamic , and between SVGD and your method, as it was done in "The promises and pitfalls of Stochastic Gradient Langevin Dynamics". Basically you can find a coupling between trajectories of your methods and the langevin dynamics and bound the wasserstein distance between the two methods. This will be insightful to see if one would mix faster with respect to the other one.

While the appendices of the paper are lengthy I don't think they explain the most selling point of the method, since they are asymptotic and there is a confusion between the time horizon for the past , and the number of particles.  if $M$ is  $\infty$ in this defintion , we are just running langevin dynamic, and not using Stein witness function. I see an important issue in this definition of the time horizon, I think this time horizon should be finite, and that from time steps less we sample N particles , and we let this $N$ go to infinity.


**Experience Assessment:**

I have published one or two papers in this area.

**Review Assessment: Checking Correctness Of Derivations And Theory:**

I assessed the sensibility of the derivations and theory.

**Review Assessment: Checking Correctness Of Experiments:**

I carefully checked the experiments.

**Review Assessment: Thoroughness In Paper Reading:**

I read the paper thoroughly.

---

> ### Author Response · Authors · 2019-11-09
> **Thanks for your constructive feedback! Below are our responses. (1/n)**
>
> Thanks very much for your comments! To make things clear, we first briefly summarize our answer regarding your questions and then we elaborate the details.
>
> Regarding your Q1: we don’t clearly get your question. In SRLD, we only maintain one single chain and thus at each time step we only have one sample. At time step K, we use samples at time $K-c_\eta, K-2c_\eta, \cdots, K-Mc_\eta$. If time step $K$ is less than $Mc_\eta$, we run Langevin. Regarding the limit of $M$, we discuss it in the response to your Q3.
>
> Regarding your Q2: We agree with you that it is better to have refined analysis on comparing mixing time. In one of our intermediate result, we show that SRLD is geometric ergodic, which indicates that its mixing would be good. However, different from Langevin, our system is with higher order and nonlinear, refined analysis comparing the mixing time is extremely challenge and we want to postpone it as future work. Compared with many published work in this area, we believe our current contribution has reached the standard of publishing. We would also want to point out that the technique/bound in your suggested paper (thanks for suggesting this paper) is not able to compare which dynamic’s mixing is faster.
>
> Regarding your Q3: As our system is nontrivial and complicate, there are some special requirements to take the limit. The way of taking limit proposed in your review comments is not appropriate. We give the correct way of taking limit below. The key idea is, in taking the limit of $M$, we also need to ensure the total number of iterations $K$ should be larger than $M$ to make SRLD not degenerate to LD. We also want to point out that, alternatively, we can obtain the M initial sample from some distribution rather than Langevin dynamics. In this case, we will no longer need to ensure $K>M$ when taking limit and our theory can be generalized to this setting very easily.
>
> Below please find our detailed answer with more information. If our answer addresses your questions, please kindly raise your score. If not, please elaborate and we will answer.

---

> ### Author Response · Authors · 2019-11-09
> **Thanks for your constructive feedback! Below are our responses. (2/n)**
>
> Q1: The definition of  $\bar{\delta}_M$ averages only M particles chosen for M time stamps. Something is off here for the continuous approximation to work at each time stamp you need N particles and then you have a past horizon M. As  $M\to\infty$ this does not matter, but I think in your implementation you are considering at each time step N particles and you average on a horizon of size $M$. Is this correct? Please clarify?
>
> A1: We are not sure whether we interpret your question correctly, we prepare the following clarifications that we guess are related to your concern. If none of the following answers correctly address your question, please let us know and we will clarify further.
>
> - We don’t use $N$ samples for each time step in our implementation. In practice, we only run one chain and thus at each time step we only have one sample. And at time step $K$, we calculate the repulsive gradient based on the past $M$ samples. Notice that here we add a thinning factor $c_\eta$ and the past M samples would be the ones at time step $K-c_\eta, K-2c_\eta,\cdots ,K-Mc_\eta$. (If K is smaller than $Mc_\eta$, in the implement we simply run Langevin). (If $K$ is larger than $Mc_\eta$, we do not use the sample at time step less than $K-Mc_\eta$). Regarding the issue on taking $M\to\infty$, please see our answer to your question 3.
>
> - The practical dynamics and discrete-time mean-field dynamics converge to each other when $M$ is sufficient large. And in the practical dynamics, we only use $M$ samples from past time steps (1 sample each time step). The key phenomenon is that the empirical distribution of the past M samples is able to approximate the mixture of marginal distributions at the past $M$ time step (which is $\tilde{\rho}_k^M$ in the main text).
>
> - In the continuous time limit, we consider how the dynamics evolves as $t$ continuously changes. Again, there is only one chain (which is just one realization on the stochastic process) and at each time step, we also only have one sample. Similar to the discrete time case, at time step $T$, given the thinning factor $c$, we calculate the repulsive gradient based on the past $M$ samples, which is the samples at time step $T-c, T-2c, \cdots, T-Mc$.
>
> Since we are not sure about whether we understand your concern correctly, let us know if we do not address your concern.

---

> ### Author Response · Authors · 2019-11-09
> **Thanks for your constructive feedback! Below are our responses. (3/n)**
>
> Q2: The theorem show only asymptotic behavior and don't quantify the intuition behind the paper , that the "coverage of the samples" is higher. Can you for instance bound the wasserstein distance between the pure langevin and Stein Repulsive dynamic , and between SVGD and your method, as it was done in “The promises and pitfalls of Stochastic Gradient Langevin Dynamics". Basically you can find a coupling between trajectories of your methods and the langevin dynamics and bound the wasserstein distance between the two methods. This will be insightful to see if one would mix faster with respect to the other one.
>
> A2: We agree with you that providing quantitative analysis on comparing mixing time would be a huge improvement. One of our intermediate result shows that the proposed SRLD is geometric ergodic (see B.2, appendix). In our assumptions, LD is also geometric ergodic. It is not clear whether SVGD is geometric ergodic (though we expect it is not and thus we may be better than SVGD). However, compared with Langevin dynamics, since our system is with higher order and nonlinear, a further refined comparison analysis on mixing time is very challenge and we believe it is out the scope of this paper and we would like to postpone it to future work.
>
> We believe our current contribution has been above the bar of publishing in this area. There are quite a few published papers proposed new sampling methods without mixing time guarantee, for example, [1, 2, 3, 4, 5].  Similar to us, they justify the proposed sampling method by showing convergence to the target distribution and support their improvement on sampling quality by some empirical evidence. Compared with many of them, we give more throughout theoretical analysis.
>
> We also would like to point out that bounding the Wasserstein distance between SRLD and LD/SVGD using the suggested way is not able to compare whose mixing is faster as it can only show whether two dynamics converges to the same limit with some extra error term (and we’ve justified that SRLD is able to approximate the target distribution). The technique in the suggested paper is for linear system and generalizing it to nonlinear and higher order case is very challenge.
>
> [1] Chen, Tianqi, Emily Fox, and Carlos Guestrin. “Stochastic gradient hamiltonian monte carlo." International conference on machine learning. 2014.
> [2] Ding, Nan, et al. "Bayesian sampling using stochastic gradient thermostats." Advances in neural information processing systems. 2014.
> [3] Şimşekli, Umut. "Fractional Langevin Monte Carlo: Exploring Levy Driven Stochastic Differential Equations for Markov Chain Monte Carlo." International Conference on Machine Learning. 2017.
> [4] Tripuraneni, Nilesh, et al. "Magnetic Hamiltonian Monte Carlo." Proceedings of the 34th International Conference on Machine Learning-Volume 70. JMLR. org, 2017.
> [5] Luo, Rui, et al. "Thermostat-assisted continuously-tempered Hamiltonian Monte Carlo for Bayesian learning." Advances in Neural Information Processing Systems. 2018.

---

> ### Author Response · Authors · 2019-11-09
> **Thanks for your constructive feedback! Below are our responses. (4/n)**
>
> Q3: While the appendices of the paper are lengthy I don't think they explain the most selling point of the method, since they are asymptotic and there is a confusion between the time horizon for the past , and the number of particles.  if  M is \infty in this defintion , we are just running langevin dynamic, and not using Stein witness function. I see an important issue in this definition of the time horizon, I think this time horizon should be finite, and that from time steps less we sample N particles , and we let this N go to infinity.
>
> A3: As we point out in addressing your last comments, the role of our theory is to justify the proposed sampling dynamics rather then comparing different dynamics. And different from most previous sampling dynamics, the proposed dynamics is nonlinear with higher order dependency on past stages, the analysis itself is challenging.
>
> Regarding the issue on $M$: Thank you for pointing out this issue! In the theory, we give non-asymptotic bound for all the approximations. Our system is actually very complicated and thus we need to be careful when taking the asymptotic limit to show the convergence. Your way of taking the limit is not appropriate. The correct way to take the limit is as follows: (we’ve also added a new remark in the main text blow theorem 4.3.)
>
> $$\lim_{k,M \to\infty, \eta \to 0^+} \mathbb{D}_{\text{BL}} (\rho_k, \rho^*)=0$$
>
> where the joint limit of $k$ and $M$ requires that $k\eta\to\infty$; $\exp(C\alpha^{2}k\eta)\eta^{2}=o(1)$; $(k\eta)/(Mc)=q(1+o(1))$ with $q>1$. Here if $q \leq 1$, we degenerate to Langevin. But when $q>1$ (intuitively that means, when $M$ is large, the number of iterations we run is larger), our dynamics is different from Langevin, which is what we do in the practice.
>
> We would also want to point out that this seemingly strange things is actually the ‘artifact’ caused by the using of Langevin dynamics to obtain $M$ initial samples: when $M \to \infty$, we need to run the Langevin for infinite long time. However, it is not really necessary to use Langevin dynamics to get $M$ initial samples ----- we can simply using some other initialization distribution and get the $M$ initial samples from that distribution (and by this setting, our dynamics is simply the second phases in Eq (3)). All our theory can be easily generalized to this setting using almost identical argument.
>
> In practice, we find it is very natural to use Langevin to obtain the initial M samples and as a burn-in process. We use this way to do all the experiments. We use the current setting to do theoretically analysis because we want our theory to be as close to practical as possible.
>
> We hope you can kindly raise your score if our answers address your concerns. Thanks!

---

### Author Response · Authors · 2019-11-09
**Summary of Revision**

We have uploaded a new version of our paper. The main differences are:

1. We added remarks in Section 3 and Section 4 to emphasize that the Langevin dynamics used in the first phase is not necessary for the second phase. Our theoretical analysis still holds for the cases that initialize $M$ samples from some initialization distribution.

2. We described a more clear asymptotic convergence result in Section 4.

3. We added higher-dimensional Gaussian and Gaussian mixture experiments in the Appendix A.2 and A.3, following the suggestions from Reviewer 1.

4. We fixed several minor typos.

---

### Decision · Program_Chairs · 2019-12-19

**Decision:**

Reject

**Comment:**

This paper proposes a new sampling mechanism which uses a self-repulsive term to increase the diversity of the samples.

The reviewers had concerns, most of which were addressed in the rebuttal. Unfortunately, none of the reviewers genuinely championed the paper. Since there were a lot of good submissions this year, we had to make decisions on the borderline papers and this lack of full support means that this submission will be rejected.

I highly encourage you to keep updating the manuscript and to rebusmit it to a later conference.